# ADVERSARIAL ATTACKS ALREADY TELL THE ANSWER: DIRECTIONAL BIAS-GUIDED TEST-TIME DEFENSE FOR VISION-LANGUAGE MODELS

**Liangsheng Liu**[1]    **Si Chen**[1]    **Jiamin Wu**[3]    **Weiwei Feng**[4,5]
**Zhixin Cheng**[1]    **Xiaotian Yin**[1]    **Wenfei Yang**[1]    **Tianzhu Zhang**[1,2,*]
[1]University of Science and Technology of China,
[2]National Key Laboratory of Deep Space Exploration, Deep Space Exploration Laboratory,
[3]The Chinese University of Hong Kong, [4]Zhejiang University, [5]Ant Group
`{liuls,cs_fisha,fengww,chengzhixin,xiaotianyin}@mail.ustc.edu.cn,`
`{yangwf, tzzhang}@ustc.edu.cn, jiaminwu@cuhk.edu.cn`

## ABSTRACT

Vision-Language Models (VLMs), such as CLIP, have shown strong zero-shot generalization but remain highly vulnerable to adversarial perturbations, posing serious risks in real-world applications. Test-time defenses for VLMs have recently emerged as a promising and efficient approach to defend against adversarial attacks without requiring costly large-scale retraining. In this work, we uncover a surprising phenomenon: under diverse input transformations, adversarial images in CLIP's feature space consistently shift along a dominant direction, in contrast to the dispersed patterns of clean images. We hypothesize that this dominant shift, termed the Defense Direction, opposes the adversarial shift, pointing features back toward their correct class centers. Building on this insight, we propose **Directional Bias-guided Defense (DBD)**, a test-time framework that estimates the Defense Direction and employs a DB-score–based two-stream reconstruction strategy to recover robust representations. Experiments on 15 datasets demonstrate that DBD not only achieves SOTA adversarial robustness while preserving clean accuracy, but also reveals the counterintuitive result that robust accuracy can even surpass clean accuracy. This demonstrates that adversarial perturbations inherently encode directional priors about the true decision boundary. Our code is available at https://github.com/liuls2002/DBD.

## 1 INTRODUCTION

Vision-Language Models (VLMs) such as CLIP (Radford et al., 2021), pre-trained on large-scale image-text pairs, enable strong cross-modal understanding and zero-shot generalization, and are now widely applied across vision and multimodal tasks (Zhang et al., 2024b). Despite its success, CLIP is highly vulnerable to adversarial perturbations: even imperceptible input distortions (Szegedy et al., 2013) can cause severe prediction errors. Such fragility poses critical safety risks in security-sensitive applications, making adversarial robustness a key challenge for reliable deployment.

Adversarial training (Madry et al., 2017; Zhang et al., 2019) is a well-studied strategy for improving model robustness. When extended to VLMs like CLIP, methods such as Adversarial Fine-Tuning (Mao et al., 2022; Wang et al., 2024; Schlarmann et al., 2024) and Adversarial Prompt Tuning (Li et al., 2024; Zhou et al., 2024) have achieved notable progress in strengthening adversarial resistance. However, these approaches rely on task-specific annotated datasets, making training costly and less accessible. Optimization on limited data may also weaken generalization and zero-shot transferability. To address these limitations, recent studies have explored test-time defenses that require no additional training, broadly categorized as prompt-based and transformation-based approaches. Prompt-based defenses (Sheng et al., 2025; Wang et al., 2025) adapt textual

---

*Corresponding author

prompts for each instance, effectively mitigating attacks but substantially increasing inference latency. Transformation-based methods, such as counterattack perturbation (Xing et al., 2025) and Gaussian noise injection (Tong et al., 2025), offer a simple and efficient way to enhance adversarial robustness by modifying inputs, yet they may degrade performance on clean images.

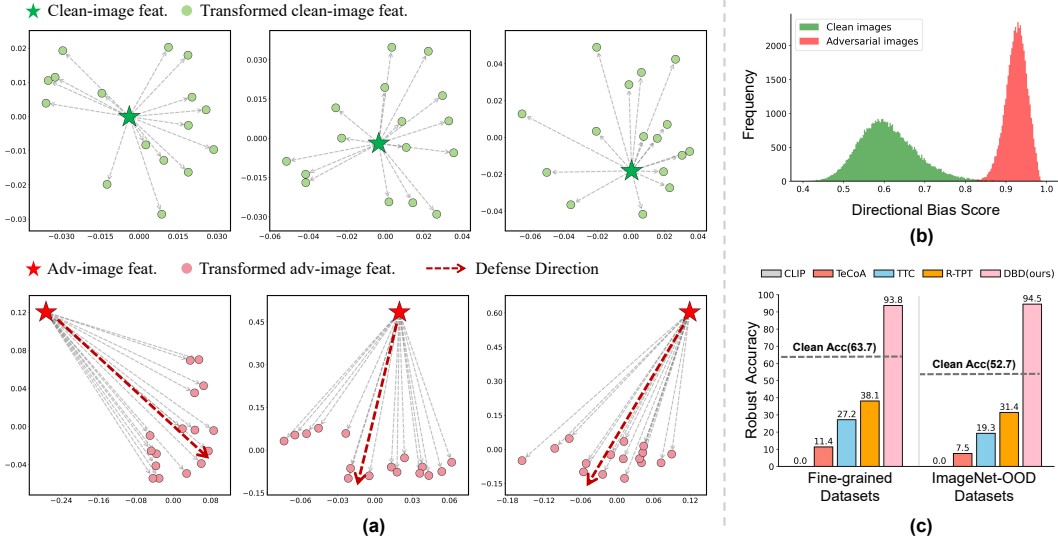

Figure 1: **Illustration of the proposed Directional Bias (DB) analysis:** (a) Visualization of image features under various transformations for clean (first row) and adversarial (second row) inputs, obtained via Multidimensional Scaling (MDS) with $1-$cosine similarity as the distance metric. Clean inputs yield dispersed feature patterns, whereas adversarial inputs exhibit strong directional bias. (b) Histogram of DB-scores on ImageNet, showing clear separation between clean and adversarial images. (c) Comparison of average robust accuracy across multiple methods on ten fine-grained datasets (left) and five ImageNet-OOD datasets (right). Our DBD consistently surpasses previous SOTA methods and even outperforms clean accuracy across all datasets. Adversarial images are generated using 100-step PGD ($\ell_\infty$, $\epsilon = 4/255$, step size $1/255$) on CLIP-ViT-B/32.

Given their effectiveness and computational efficiency, transformation-based methods have emerged as a promising approach for test-time defense. Prior studies (Guo et al., 2017; Cohen et al., 2019; Dziugaite et al., 2016; Xie et al., 2017) have shown that various image transformations can mitigate adversarial effects. However, the underlying mechanism of how and why the adversarial effects are alleviated remains unexplored, limiting further progress in defense design. To address this, we analyze the latent feature space to investigate how diverse transformations alter image features and thereby mitigate adversarial effects. As shown in Fig. 1(a), when applying various transformations to an input image, the transformed features of a clean image tend to scatter around the original feature, whereas those of an adversarial image consistently shift toward a specific direction, presenting a skewed pattern. To quantify this phenomenon, we further introduce a **Directional Bias (DB) score** to measure the directional concentration of transformed features. As shown in Fig. 1(b), the DB-score exhibits a clear bimodal distribution, effectively distinguishing adversarial from clean ones, as adversarial images consistently exhibit high and concentrated scores.

The above observation prompts a key question: *what does the direction of transformed features represent?* Recall that adversarial attacks work by shifting features away from original class centers, thereby inducing misclassification. We therefore hypothesize that this dominant direction could be anti-parallel to the adversarial shift, pointing features back toward their correct class centers. Building on this insight, we propose **Directional Bias-guided Defense (DBD)**, a test-time framework for VLMs that leverages this specific direction, referred to as **Defense Direction**, to uncover discriminative features. To capture robust Defense Direction, DBD applies a wide range of transformations across spatial, pixel, and frequency domains to obtain diverse augmented features, and then uses entropy-based filtering to retain high-quality ones. Leveraging the DB-score to distinguish between adversarial and clean inputs, we propose a two-stream feature reconstruction strategy to enhance test-time defense: for high DB-score examples, adversarial features are linearly shifted along the

Defense Direction to restore correct representations; while for low DB-score examples, the average transformed features are used as test-time augmentation for stabilizing representations.

We conduct extensive experiments across ten fine-grained classification datasets and five ImageNet-OOD datasets. The results demonstrate that our method not only preserves performance on clean images but also achieves substantial improvements over previous state-of-the-art defenses on adversarial examples across all datasets. Remarkably, as shown in Fig. 1(c), the classification accuracy on adversarial images even surpasses that on clean images. *This counterintuitive result justifies that the generation of adversarial examples guided by ground-truth labels implicitly encodes directional priors about the true decision boundary, which we exploit to achieve effective defense.*

Our main contributions are as follows: (1) To the best of our knowledge, we are the first to show that adversarial perturbations implicitly encode directional priors of the true decision boundary, which can be reliably estimated using multiple transformations. (2) We propose Directional Bias-guided Defense (DBD), a test-time framework that leverages these directional priors through Defense Direction estimation and a two-stream reconstruction strategy based on the proposed DB-score, enabling effective and efficient defense. (3) We validate DBD on 15 datasets, demonstrating superior adversarial robustness while preserving zero-shot performance on clean images. In some cases, our method even surpasses the performance on clean images when evaluated on adversarial images.

## 2 RELATED WORKS

**Vision-Language Models (VLMs).** CLIP (Radford et al., 2021), trained on large-scale image-text pairs, has become a cornerstone vision-language model (VLM) with strong zero-shot generalization and cross-modal reasoning (Zhang et al., 2024b). Building on this paradigm, ALIGN (Jia et al., 2021) and BLIP-2 (Li et al., 2023) further scale or refine the alignment of image-text pairs, while LLaVA (Liu et al., 2023) extends VLMs toward instruction-following and conversational tasks. By aligning modalities in a shared embedding space, these models provide powerful task-agnostic representations. However, prior studies (Zhao et al., 2023; Schlarmann & Hein, 2023) have shown that VLMs are highly vulnerable to adversarial attacks, posing a critical barrier to their deployment in safety-sensitive applications.

**Adversarial Attacks and Defenses.** Adversarial perturbations are small but carefully crafted input distortions that can drastically mislead deep neural networks (Szegedy et al., 2013; Feng et al., 2021; 2023; 2024a;b). Early works proposed gradient-based attacks such as FGSM (Goodfellow et al., 2014), iterative methods like PGD (Madry et al., 2017), and optimization-based approaches such as CW (Carlini & Wagner, 2017). More recent efforts have introduced adaptive attacks such as AutoAttack (AA) (Croce & Hein, 2020b), a robust benchmark combining four attacks: the score-based black-box Square (Andriushchenko et al., 2020), the minimal-$\ell_p$-perturbation FAB (Croce & Hein, 2020a), APGD-CE (using cross-entropy loss), and APGD-DLR (using difference-of-logits-ratio loss). To counter adversarial threats, defenses have been extensively explored. Adversarial training (Madry et al., 2017; Zhang et al., 2019; Shafahi et al., 2019; Wong et al., 2020) optimizes models on perturbed examples to enhance robustness. Input purification (Guo et al., 2017; Xie et al., 2017) transforms inputs toward the clean distribution. Recent diffusion-based purification methods (Nie et al., 2022; Chung et al., 2022) show promise in removing perturbations but often incur high computational cost.

**Adversarial Robustness of VLMs.** For VLMs such as CLIP, several extensions of adversarial training (Mao et al., 2022; Wang et al., 2024; Schlarmann et al., 2024; Li et al., 2024; Zhou et al., 2024) have been proposed to enhance robustness. TeCoA (Mao et al., 2022) examines the effect of fine-tuning and visual prompt tuning on the zero-shot adversarial robustness of VLMs. Adversarial Prompt Tuning methods, including APT (Li et al., 2024) and AdvPT (Zhang et al., 2024a), focus on optimizing textual prompts without modifying model parameters. However, these methods rely on annotated data and may weaken generalization, motivating test-time defenses that require no additional training. Prompt-based test-time methods such as R-TPT (Sheng et al., 2025) and TAPT (Wang et al., 2025) adapt prompts on a per-instance basis, achieving reasonable robustness at the cost of substantial inference overhead. Transformation-based methods mitigate adversarial attacks by modifying input images. For example, TTC (Xing et al., 2025) generates counterattack perturbations for adversarial images, and AOM (Tong et al., 2025)injects Gaussian noise into inputs. These approaches can improve robustness in practice but often degrade performance on clean

images. Our method exploits directional bias in latent feature space to reconstruct features under diverse transformations, enhancing adversarial robustness while preserving clean performance, and achieving an efficient balance between robustness and computational cost.

# 3 METHOD

## 3.1 PRELIMINARIES

**Zero-shot classification of CLIP.** CLIP (Radford et al., 2021) is a VLM that projects images and texts into a shared embedding space and measures their relationships using cosine similarity. For zero-shot classification, CLIP consists of two pre-trained encoders: a visual encoder $\mathcal{E}_v$ and a text encoder $\mathcal{E}_t$. For an $C$-class classification task, given an image $x_{test}$ and a set of class names with prompts $T_c, c \in [1, C]$, CLIP computes text features: $\boldsymbol{f}_{t_c} = \mathcal{E}_t(T_c)$, for each class $c$, and image feature $\boldsymbol{f}_v = \mathcal{E}_v(x_{test})$. The prediction probability for class $c$ is calculated as:

$$P_{\text{CLIP}}(y = c \mid x_{\text{test}}) = \frac{\exp(\cos(\boldsymbol{f}_{t_c}, \boldsymbol{f}_v)/t)}{\sum_{c'=1}^{C} \exp(\cos(\boldsymbol{f}_{t_{c'}}, \boldsymbol{f}_v)/t)}, \tag{1}$$

where $\cos(\cdot, \cdot)$ is the cosine similarity between the features, and $t$ is a temperature parameter that controls the sharpness of the distribution. The final classification decision is determined by selecting the class with the highest probability:

$$\hat{y} = \arg\max_{c \in [1,C]} P_{\text{CLIP}}(y = c \mid x_{\text{test}}), \tag{2}$$

where $\hat{y}$ represents the predicted class label.

**Adversarial attacks for CLIP.** Despite its strong zero-shot performance, CLIP is particularly sensitive to small adversarial perturbations (Szegedy et al., 2013). Following recent SOTA test-time method R-TPT (Sheng et al., 2025), we consider a threat model where the attacker has full access to the vanilla CLIP model, but no knowledge of the defense mechanism. This reflects real-world deployment: foundation models like CLIP have publicly available weights, while test-time defenses are typically deployed privately. In this setting, adversarial examples are crafted against CLIP as:

$$\delta = \arg\max_{\delta'} \mathcal{L}(\text{CLIP}(x + \delta', T), y), \quad \text{s.t. } \|\delta'\|_p \leq \epsilon, \tag{3}$$

where $y$ is the ground-truth label of input image $x$, $T$ is a set of class names with prompts, $\mathcal{L}$ is a loss function (typically cross-entropy loss), and $\epsilon$ is the attack budget controlling the magnitude of perturbations to remain imperceptible.

## 3.2 DBD FOR VLMS

We propose Directional Bias-guided Defense (DBD), a test-time framework for defending VLMs against adversarial attacks. The core idea of DBD is to leverage multiple input transformations to construct diverse reference features, analyze their directional bias relative to the original features, and use this property to guide feature reconstruction. The overall framework consists of three main components: input transformations & feature filtering, Directional Bias (DB) computation, and two-stream feature reconstruction, as illustrated in Fig. 2.

**Image Transformations & Feature Filtering.** Since individual transformations have inherent drawbacks, relying on a single transformation may produce unreliable features. For example, random cropping is stochastic and may capture mostly background, while filtering can excessively blur important details. To improve robustness, we apply a diverse set of transformations to generate multiple feature candidates, which leverages complementary strengths across transformations to both preserve task-relevant information and disrupt adversarial noise.

We construct an image transformation library covering diverse transformations across three domains: (1) *Spatial domain*: including random cropping, scaling, and flipping. These geometric

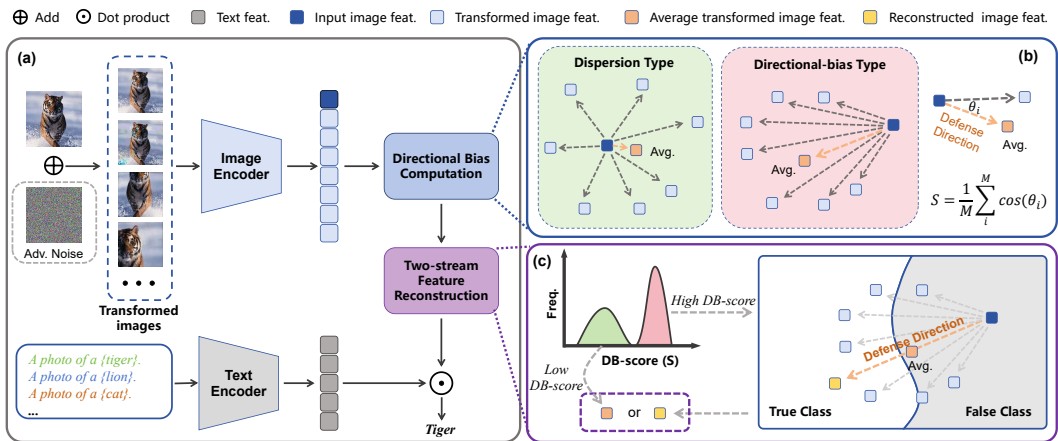

Figure 2: **Overview of the proposed Directional Bias-guided Defense (DBD).** (a) Framework: multiple transformations are applied to the input image, and high-quality transformed features are retained by entropy-based filtering; then the Defense Direction and DB-score are computed for feature reconstruction and classification. (b) Directional Bias (DB) Computation: the Defense Direction is defined from the original feature to the average transformed feature, and the DB-score is the mean cosine similarity between the Defense Direction and each individual displacement vector. (c) Two-stream Feature Reconstruction: for high DB-score (likely adversarial images), the original features are shifted further along the Defense Direction to obtain more robust representations; for low DB-score (likely clean images), the average transformed features are used as test-time augmentation for stabilizing representations.

operations alter object position, size, and orientation, thereby disrupting the structured alignment of adversarial perturbations and weakening their effect. (2) *Pixel domain*: including bit-depth compression (quantization), JPEG compression-decompression, and additive Gaussian noise. These pixel-level modifications distort or overwrite fine-grained perturbations, making them less effective in misleading the model. (3) *Frequency domain*: including Gaussian, mean, and median filtering. By smoothing or suppressing high-frequency components, these filters reduce adversarial noise while largely preserving the semantic content of the image.

Transformed features exhibit varying quality across different transformations, so we apply a feature filtering step to select the most informative and reliable representations. Following common practice in test-time adaptation (Shu et al., 2022; Chen et al., 2025), we use the entropy of the model's prediction as a quality metric. Specifically, given $n$ transformed images, we pre-compute their features $\boldsymbol{f}_i(i = 1, 2, ..., n)$ and classification probabilities with CLIP, then calculate the entropy for each:

$$E_i = -\sum_c p_{i,c} \log p_{i,c}, \tag{4}$$

where $p_{i,c}$ denotes the predicted probability of the $i$-th transformed image feature for class $c$. We then select the $k$ transformed image features with the lowest entropy as high-quality features for subsequent processing:

$$\mathcal{F}_{\mathrm{ref}} = \{\boldsymbol{f}_i \mid i \in \mathcal{I}_k\}, \quad \mathcal{I}_k = \underset{i \subset \{1,...,n\}, |i|=k}{\arg\min} \sum_{j \in i} E_j. \tag{5}$$

**Directional Bias (DB) Computation.** After applying multiple transformations and feature filtering, we obtain a set of high-quality transformed image features. Visualization (Fig. 1(a)) shows that for clean images, the transformed features exhibit a dispersed pattern around the original feature. In contrast, for adversarial inputs, they consistently shift toward a specific direction, presenting a directional bias pattern. This occurs because transformations partially mitigate adversarial perturbations, aligning the features closer to the clean feature distribution.

Given a set of transformed features $\mathcal{F}_{\mathrm{ref}} = \{\boldsymbol{f}_i \mid i = 1, \ldots, k\}$ and the original feature $\boldsymbol{f}_0$, we define the direction vectors for each transformed feature $\boldsymbol{f}_i$ as unit vectors $\boldsymbol{d}_i = (\boldsymbol{f}_i - \boldsymbol{f}_0)/\|\boldsymbol{f}_i - \boldsymbol{f}_0\|_2$,

and compute the Defense Direction $\boldsymbol{d}_{\mathrm{def}}$ as

$$\bar{\boldsymbol{d}} = \frac{1}{k} \sum_{i=1}^{k} (\boldsymbol{f}_i - \boldsymbol{f}_0), \quad \boldsymbol{d}_{\mathrm{def}} = \frac{\bar{\boldsymbol{d}}}{\|\bar{\boldsymbol{d}}\|_2}, \tag{6}$$

where $\| \cdot \|_2$ denotes the $\ell_2$ norm (Euclidean distance). The DB-score is computed as the average cosine similarity between each direction and the Defense Direction:

$$S_{\mathrm{db}} = \frac{1}{k} \sum_{i=1}^{k} \langle \boldsymbol{d}_i, \boldsymbol{d}_{\mathrm{def}} \rangle, \tag{7}$$

where $\langle \cdot, \cdot \rangle$ denotes the inner product. As shown in Fig. 1(b), the DB-score exhibits a clear bimodal distribution for clean and adversarial images, allowing simple thresholding to separate them.

**Two-stream Feature Reconstruction.** Adversarial perturbations shift image features away from the true classification region, thereby inducing misclassification. Intuitively, the Defense Direction $\boldsymbol{d}_{\mathrm{def}}$ could be anti-parallel to the adversarial shift, pointing features back toward their correct classification region. Guided by this intuition, we reconstruct more robust features by linearly shifting the input feature along the Defense Direction. However, due to stochasticity or imperfections in the transformations, the computed Defense Direction may be inaccurate. To reduce the negative impact of inaccurate Defense Direction, we propose a two-stream reconstruction strategy based on the DB-score: (1) High DB-score stream: a high $S_{db}$ indicates a likely adversarial image and a more reliable Defense Direction. In this case, we shift the feature along the Defense Direction to enhance its distinction from the original. (2) Low DB-score stream: a low $S_{db}$ suggests a likely clean image with less reliable direction. For these examples, we use the average of transformed features as test-time augmentation for stabilizing representations. Formally, we introduce a threshold $\tau$ on DB-score $S_{db}$:

$$\hat{\boldsymbol{f}} = \boldsymbol{f}_0 + l \cdot \boldsymbol{d}_{\mathrm{def}}, \quad l = \begin{cases} \|\bar{\boldsymbol{d}}\|_2, & S_{db} \leq \tau \\ \lambda \cdot \|\bar{\boldsymbol{d}}\|_2, & S_{db} > \tau \end{cases}, \tag{8}$$

where $\hat{\boldsymbol{f}}$ is the reconstructed feature and $\lambda$ is a hyperparameter controlling the magnitude of the feature shift. In practice, we use the distance from the average transformed features to the original feature $\|\bar{\boldsymbol{d}}\|_2$ as a reference for the shift magnitude, which is then scaled by $\lambda$. Finally, we use the reconstructed image feature $\hat{\boldsymbol{f}}$ to compute the predicted classification result, as given in Eq.1. Our DBD is summarized in Algorithm 1 in the appendix.

# 4 EXPERIMENTS

## 4.1 EXPERIMENT SETUP

**Datasets.** Following prior works (Sheng et al., 2025; Li et al., 2024) on the adversarial robustness of CLIP, we evaluate our proposed test-time DBD on ten fine-grained classification datasets and five ImageNet-based out-of-distribution(OOD) benchmarks. The fine-grained datasets span diverse domains: general objects (*Caltech101* (Fei-Fei et al., 2004)), animals (*Pets* (Parkhi et al., 2012)), plants (*Flower102* (Nilsback & Zisserman, 2008)), vehicles (*Cars* (Krause et al., 2013), *Aircraft*) (Maji et al., 2013), textures (*DTD* (Cimpoi et al., 2014)), satellite imagery (*EuroSAT* (Helber et al., 2019)), human actions (*UCF101* (Soomro et al., 2012)), scenes (*SUN397* (Xiao et al., 2010)), and food (*Food101* (Bossard et al., 2014)). For ImageNet-OOD evaluation, we use *ImageNet* (Deng et al., 2009) and four established variants: *ImageNet-A* (Hendrycks et al., 2021b), *ImageNet-V2* (Recht et al., 2019), *ImageNet-R* (Hendrycks et al., 2021a), and *ImageNet-S* (Wang et al., 2019). Since our method targets test-time adversarial robustness, we do not require access to any training sets.

**Implementation details.** We use official pre-trained CLIP backbones (ResNet-50 (He et al., 2016), ViT-B/32, and ViT-B/16 (Dosovitskiy et al., 2020)) as the base models. Adversarial images are generated with PGD (Madry et al., 2017) under the $L_\infty$ norm constraint. Following prior works (Sheng et al., 2025; Li et al., 2024), we evaluate two threat levels. For low-strength attack, we use PGD-10 with $\epsilon = 1/255$ on CLIP-ResNet50; for high-strength attack, we use PGD-100 with $\epsilon = 4/255$ on CLIP-ViT-B/32 and CLIP-ViT-B/16. The step size for all attacks is $\alpha = \epsilon/4$. For our DBD, we

apply $n = 31$ transformations per input, yielding 32 images including the original, and then select $k = 16$ transformed image features via entropy-based filtering. The DB-score threshold is $\tau = 0.8$, and the feature shift magnitude is set to $\lambda = 2.5$. Both are estimated from ImageNet validation set (50k images). Experiments are conducted in PyTorch on RTX 3090 GPUs.

**Baselines.** We compare DBD with several existing methods, including adversarial fine-tuning on ImageNet (TeCoA (Mao et al., 2022)), adversarial prompt tuning on downstream datasets with 16 shots (APT (Li et al., 2024)), test-time prompt tuning (R-TPT (Sheng et al., 2025)), test-time input transformation method (TTC (Xing et al., 2025)), and the original CLIP (Radford et al., 2021) models. Except for APT, which uses few-shot tuning , all other methods operate in a zero-shot setting. Baseline results are obtained from official reports or reproduced using official code.

Table 1: Results (%) of clean accuracy (Acc.) and robust accuracy (Rob.) of various defense methods on ten **fine-grained classification datasets**. Robust accuracies are highlighted with gray background . Best clean accuracies are (**bold**), and best robust accuracies are (**bold red**).

| Method | Caltech101 Acc. | Rob. | Pets Acc. | Rob. | Cars Acc. | Rob. | Flower102 Acc. | Rob. | Aircraft Acc. | Rob. | DTD Acc. | Rob. | EuroSAT Acc. | Rob. | UCF101 Acc. | Rob. | SUN397 Acc. | Rob. | Food101 Acc. | Rob. | Avg. Acc. | Rob. |
|---|---|---|---|---|---|---|---|---|---|---|---|---|---|---|---|---|---|---|---|---|---|---|
| *PGD-10 ($\epsilon = 1/255$) on CLIP-ResNet50* | | | | | | | | | | | | | | | | | | | | | | |
| CLIP | 89.1 | 2.1 | 85.0 | 0.0 | 57.3 | 0.0 | 65.9 | 0.0 | 19.6 | 0.0 | 48.5 | 0.4 | 37.5 | 0.0 | 59.7 | 0.0 | 62.7 | 0.0 | **75.6** | 0.0 | 60.1 | 0.3 |
| TeCoA | 78.2 | 64.1 | 76.2 | 54.4 | 24.1 | 9.2 | 32.6 | 17.3 | 6.6 | 2.4 | 30.7 | 21.4 | 23.8 | 19.0 | 40.4 | 21.8 | 38.6 | 19.7 | 29.2 | 12.3 | 38.1 | 24.2 |
| R-TPT | 86.0 | 79.9 | 84.7 | 73.4 | 58.4 | 42.1 | 60.7 | 51.0 | 18.1 | 12.3 | 44.1 | 34.3 | 21.2 | 15.8 | 59.2 | 50.3 | 60.8 | 50.7 | 73.3 | 57.8 | 56.3 | 46.8 |
| DBD | **90.1** | 98.7 | **86.0** | 95.9 | **60.0** | 86.2 | 65.9 | 88.3 | 21.6 | 56.3 | 47.9 | 85.2 | 29.4 | 81.3 | 60.6 | 88.9 | 63.8 | 93.2 | 75.0 | 97.4 | 60.0 | 87.1 |
| *PGD-100 ($\epsilon = 4/255$) on CLIP-ViT-B/32* | | | | | | | | | | | | | | | | | | | | | | |
| CLIP | 93.3 | 0.1 | 86.6 | 0.0 | 61.2 | 0.0 | 67.0 | 0.0 | 20.6 | 0.0 | 49.9 | 0.0 | 50.8 | 0.0 | 63.6 | 0.0 | 65.7 | 0.0 | 78.7 | 0.0 | 63.7 | 0.0 |
| TeCoA | 81.5 | 46.1 | 64.4 | 16.7 | 11.5 | 1.1 | 30.1 | 9.5 | 6.7 | 0.6 | 29.3 | 12.7 | 13.8 | 11.1 | 34.0 | 6.3 | 34.7 | 6.5 | 22.4 | 3.0 | 32.8 | 11.4 |
| APT | 86.6 | 57.6 | 66.6 | 17.2 | 41.9 | 9.9 | **84.4** | 47.0 | **28.7** | 6.8 | 47.5 | 21.4 | **67.2** | 23.5 | 58.2 | 18.9 | 46.6 | 10.5 | 33.3 | 6.8 | 56.1 | 22.0 |
| TTC | 89.5 | 47.6 | 61.0 | 41.5 | 45.9 | 21.3 | 65.5 | 29.2 | 15.4 | 11.1 | 39.5 | 20.4 | 44.8 | 15.4 | 63.1 | 44.2 | 64.0 | 44.1 | 74.4 | 43.5 | 60.2 | 38.1 |
| R-TPT | 91.0 | 77.8 | 84.8 | 57.7 | 63.3 | 28.3 | 63.3 | 38.8 | 19.6 | 10.1 | 42.7 | 29.9 | 31.9 | 6.5 | 63.1 | 44.2 | 64.0 | 44.1 | 78.5 | 43.5 | 60.2 | 38.1 |
| DBD | **93.8** | 99.0 | **86.8** | 96.2 | 63.7 | 91.2 | 68.7 | 94.7 | 22.4 | 66.3 | 51.5 | 88.3 | 41.7 | 92.6 | 65.3 | 92.2 | 67.1 | 94.2 | 80.1 | 98.4 | 64.1 | 91.3 |
| *PGD-100 ($\epsilon = 4/255$) on CLIP-ViT-B/16* | | | | | | | | | | | | | | | | | | | | | | |
| CLIP | 94.2 | 0.0 | 90.3 | 0.0 | 66.2 | 0.0 | 73.0 | 0.0 | 27.1 | 0.0 | 53.2 | 0.0 | **55.7** | 0.0 | 67.0 | 0.0 | 67.9 | 0.0 | 84.2 | 0.0 | **67.9** | 0.0 |
| TTC | 90.3 | 16.1 | 57.9 | 17.7 | 57.4 | 11.3 | 68.5 | 19.8 | 21.7 | 2.8 | 41.5 | 15.3 | 44.7 | 0.6 | 64.8 | 4.9 | 51.5 | 15.7 | 81.0 | 21.2 | 58.0 | 12.5 |
| R-TPT | 93.7 | 83.1 | 87.4 | 63.3 | 67.0 | 36.0 | 68.1 | 46.4 | 24.0 | 14.4 | 46.6 | 34.9 | 34.6 | 10.2 | 67.8 | 47.3 | 65.7 | 46.5 | 84.3 | 49.7 | 63.9 | 43.2 |
| DBD | **94.8** | 99.4 | **90.7** | 97.1 | 67.8 | 93.4 | 73.6 | 97.5 | 29.7 | 71.9 | 54.5 | 92.3 | 44.9 | 93.3 | 68.2 | 95.9 | 69.0 | 97.4 | 84.7 | 99.5 | 67.8 | 93.8 |

Table 2: Results (%) of clean accuracy (Acc.) and robust accuracy (Rob.) of various defense methods on five **ImageNet-OOD datasets**. Robust accuracies are highlighted with gray background . Best clean accuracies are (**bold**), and best robust accuracies are (**bold red**).

| Attack & Model | Method | ImageNet Acc. | Rob. | ImageNet-A Acc. | Rob. | ImageNet-V2 Acc. | Rob. | ImageNet-R Acc. | Rob. | ImageNet-S Acc. | Rob. | Avg. Acc. | Rob. |
|---|---|---|---|---|---|---|---|---|---|---|---|---|---|
| PGD-10 ($\epsilon = 1/255$) on CLIP-ResNet50 | CLIP | 61.5 | 0.0 | 23.8 | 0.0 | 54.7 | 0.0 | 60.0 | 0.4 | 35.6 | 0.3 | 47.1 | 0.2 |
| | TeCoA | 48.4 | 28.1 | 4.9 | 1.2 | 39.6 | 21.7 | 40.6 | 25.7 | 18.3 | 11.6 | 30.3 | 17.7 |
| | R-TPT | 60.8 | 47.3 | **28.0** | 14.2 | 54.7 | 41.6 | 57.7 | 46.6 | 34.0 | 26.0 | 47.0 | 35.1 |
| | DBD | **63.1** | 94.5 | 23.0 | 89.0 | **55.7** | 93.0 | **63.0** | 94.0 | **38.1** | 73.7 | **48.6** | 88.8 |
| PGD-100 ($\epsilon = 4/255$) on CLIP-ViT-B/32 | CLIP | 64.4 | 0.0 | 31.1 | 0.0 | 57.2 | 0.0 | 68.3 | 0.0 | 42.5 | 0.0 | 52.7 | 0.0 |
| | TeCoA | 39.5 | 9.7 | 4.2 | 0.3 | 32.4 | 7.3 | 38.0 | 12.7 | 18.5 | 7.6 | 26.5 | 7.5 |
| | TTC | 35.4 | 25.7 | 27.5 | 8.4 | 41.3 | 21.4 | 53.4 | 28.1 | 29.8 | 12.8 | 37.5 | 19.3 |
| | R-TPT | 64.2 | 40.4 | **36.6** | 11.0 | 58.0 | 34.3 | 70.0 | 47.9 | 41.7 | 23.6 | 54.1 | 31.4 |
| | DBD | **66.3** | 95.3 | 32.9 | 95.3 | **59.1** | 94.3 | **71.7** | 98.2 | **45.1** | 89.5 | **55.0** | 94.5 |
| PGD-100 ($\epsilon = 4/255$) on CLIP-ViT-B/16 | CLIP | 69.6 | 0.0 | 50.6 | 0.0 | 63.4 | 0.0 | 77.1 | 0.0 | 49.1 | 0.0 | 62.0 | 0.0 |
| | TTC | 37.8 | 17.4 | 46.6 | 9.9 | 48.9 | 16.1 | 63.3 | 12.4 | 38.5 | 1.9 | 47.0 | 11.5 |
| | R-TPT | 69.4 | 46.6 | **57.9** | 20.7 | 63.9 | 40.2 | 77.0 | 57.6 | 47.9 | 30.3 | 63.2 | 39.1 |
| | DBD | **71.1** | 97.7 | 52.1 | 98.9 | **64.7** | 97.3 | **79.5** | 99.4 | **51.1** | 95.3 | **63.7** | 97.7 |

## 4.2 Main Results

**Results on fine-grained datasets.** We evaluate the adversarial robustness of DBD on ten fine-grained classification datasets, with results summarized in Table 1. Under the CLIP-ViT-B/32 and PGD-100 ($\epsilon = 4/255$) setting, the original CLIP model demonstrates strong zero-shot classification performance (63.7%) but is almost entirely vulnerable to adversarial attacks (0.0%). Adversarial fine-tuning (TeCoA) improves defense to 11.4%, but at the cost of reduced zero-shot performance on clean images (63.7% → 32.8%). Adversarial prompt tuning (FAP) further enhances robustness (0.0% → 22.0%) while mitigating the drop in clean accuracy (63.7% → 56.1%), though its few-shot setting introduces data dependency. Test-time input transformation method (TTC) achieves moderate robustness on CLIP-ViT-B/32 (0.0% → 27.2%) but are sensitive to model architecture, with only 12.5% on CLIP-ViT-B/16. Previous state-of-the-art test-time prompt tuning method (R-TPT)

attains better robustness ($0.0\% \rightarrow 38.1\%$) and largely preserves zero-shot performance on clean images ($63.7\% \rightarrow 60.2\%$). In contrast, ours DBD maintains zero-shot performance close to the original CLIP across all three backbones (even exceeding it on CLIP-ViT-B/32) and achieves substantially higher robustness on adversarial examples, reaching 93.8% on CLIP-ViT-B/16, significantly surpassing both R-TPT and the performance on clean examples.

**Results on ImageNet-OOD datasets.** Results on ImageNet-OOD benchmarks are summarized in Table 2. The original CLIP model demonstrates strong robustness to distribution shifts but remains highly vulnerable to adversarial attacks. Notably, our DBD method not only preserves but slightly improves zero-shot classification on clean images (e.g., $52.7\% \rightarrow 55.0\%$ on CLIP-ViT-B/32), which we attribute to the combination of diverse image transformations and entropy-based feature filtering that produces more robust and accurate features. Across all three backbones, DBD substantially outperforms the previous state-of-the-art R-TPT on adversarial examples, achieving up to 97.7% on CLIP-ViT-B/16, a performance that even surpasses the classification accuracy on clean examples.

**Discussion.** *Remarkably, the classification accuracy on adversarial images significantly exceeds that on clean images. This counterintuitive result suggests that the generation of adversarial examples guided by ground-truth labels implicitly encodes directional priors about the true decision boundary. Our method leverages this by applying multiple image transformations to estimate the Defense Direction , then linearly shifting features along it to reconstruct robust representations.*

## 4.3 MORE ANALYSIS

Table 3: Robust accuracy (%) under PGD-100 ($\epsilon = 4/255$) on CLIP-ViT-B/16 using **pseudo-labels** across six fine-grained datasets. The last row shows the clean accuracy as a reference.

| Method | Caltech101 | Pets | Flower102 | Aircraft | DTD | UCF101 | Avg. |
|--------|-----------|------|-----------|----------|------|--------|------|
| CLIP   | 1.7       | 3.1  | 2.7       | 1.5      | 4.2  | 4.5    | 2.9  |
| R-TPT  | 84.1      | 66.1 | 49.8      | 18.2     | 37.1 | 52.7   | 51.3 |
| DBD    | **94.1**  | **88.6** | **72.3** | **26.3** | **53.1** | **66.5** | **66.8** |
| Clean  | 94.2      | 90.3 | 73.0      | 27.2     | 53.2 | 67.0   | 67.5 |

**Analysis under PGD attack with pseudo-label.** The previous experiments demonstrate that when PGD generates adversarial examples using ground-truth labels, our method achieves robust accuracy which significantly exceeds clean accuracy. To further validate its effectiveness without relying on ground-truth labels, we consider a pseudo-label setting. Specifically, we use CLIP's own predictions on clean images as pseudo-labels to guide PGD in generating adversarial examples. As shown in Table 3, under this setting the vanilla CLIP model remains highly vulnerable. In contrast, our method consistently outperforms R-TPT and achieves robust accuracy comparable to clean accuracy (e.g., $67.5\% \rightarrow 66.8\%$ on CLIP-ViT-B/16). This indicates that our method is able to fully leverage the directional prior information carried by adversarial examples.

**Analysis under various attacks.** To demonstrate the generality of our method, we evaluate DBD and baseline methods under additional adversarial attacks, including FGSM (Goodfellow et al., 2014), CW (Carlini & Wagner, 2017), AutoAttack (AA) (Croce & Hein, 2020b), and four component attacks of AA: Square Attack (Andriushchenko et al., 2020), targeted FAB (Croce & Hein, 2020a), untargeted APGD-CE, and targeted APGD-DLR. We evaluate various attacks on the same six fine-grained datasets in Table 3, with the results summarized in Table 4. DBD consistently demonstrates robust defense performance across all attack types, significantly outperforming R-TPT. Notably, under AA on CLIP-ViT-B/16, DBD achieves an average robust accuracy of 69.8%, surpassing the average clean accuracy of 67.5%.

Table 4: Average robust accuracy (%) across six fine-grained datasets under **various attacks** on CLIP-ViT-B/16. All attacks are conducted under the $\ell_\infty$ norm with perturbation budget $\epsilon = 4/255$.

| Method | FGSM | CW | AA | FAB | Square | APGD-CE | APGD-DLR | Avg. |
|--------|------|------|------|------|--------|---------|----------|------|
| CLIP   | 11.8 | 1.0  | 0.0  | 12.9 | 11.7   | 0.1     | 0.1      | 5.1  |
| R-TPT  | 38.9 | 55.6 | 45.0 | 59.5 | 59.2   | 45.0    | 48.8     | 50.4 |
| DBD    | **75.2** | **69.1** | **69.8** | **69.3** | **65.1** | **69.8** | **68.5** | **69.2** |

**Geometric Verification of the Proposed Defense Direction.** To validate our hypothesis that the Defense Direction genuinely points toward the correct decision region, we conduct a geometric analysis by measuring the cosine similarity between the estimated Defense Direction and two critical reference directions. The first is the *Clean Direction*, defined as the vector from the adversarial image's feature to its corresponding clean (unperturbed) counterpart's feature. The second is the *Class Centroid Direction*, defined as the vector from the adversarial image's feature to the centroid of features belonging to the true class (computed using correctly classified clean images).

Table 5 presents the average cosine similarities across multiple fine-grained datasets under the PGD-100 ($\epsilon$=4/255) attack on CLIP-ViT-B/16. The results demonstrate that our proposed Defense Direction exhibits extremely high similarity ($\approx 0.95$) with the Clean Direction and substantial similarity ($\approx 0.90$) with the Class Centroid Direction. These high cosine similarities provide strong geometric evidence that the Defense Direction indeed aligns closely with both the clean feature direction and the correct class centroid, supporting our hypothesis that the Defense Direction points back toward the correct decision region.

Table 5: Average cosine similarity between Defense Direction and reference directions across datasets under PGD-100($\epsilon = 4/255$) attack on CLIP-ViT-B/16.

| Reference Direction | Pets | Caltech101 | Food101 | Cars | ImageNet |
|---|---|---|---|---|---|
| Clean Direction | 0.957 | 0.945 | 0.939 | 0.943 | 0.951 |
| Class Centroid Direction | 0.932 | 0.917 | 0.898 | 0.905 | 0.892 |

## 4.4 ABLATION STUDY

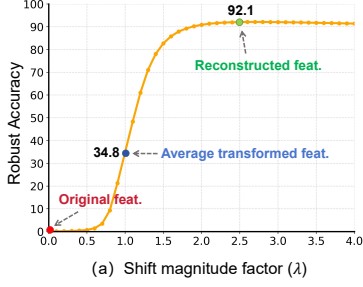 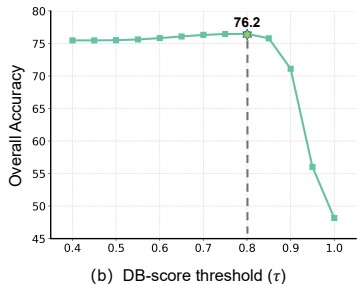

(a) Shift magnitude factor ($\lambda$)          (b) DB-score threshold ($\tau$)

Figure 3: (a) Average robust accuracy (%) across 15 datasets under three attack settings for different values of the feature shift magnitude factor $\lambda$. (b) Average overall accuracy (%) for different values of the DB-score threshold $\tau$, computed by averaging both clean and robust accuracies across all datasets and attack settings.

**Ablation of the feature shift magnitude factor $\lambda$.** To evaluate the impact of the shift magnitude factor $\lambda$ on the performance of DBD, we test three attack settings across 15 datasets. The average robust accuracy of all is shown in Fig.3(a). When $\lambda = 0$, the reconstructed feature reduces to the original feature, yielding nearly no defense. With $\lambda = 1.0$, the reconstructed feature becomes the average of transformed features, achieving an average robust accuracy of 34.8%. At $\lambda = 2.5$, DBD reaches 92.1% accuracy, demonstrating that linearly shifting the original feature along the Defense Direction reconstructs robust features that align with the correct class. This highlights the reliability of the Defense Direction identified by DBD.

**Ablation of the DB-score threshold $\tau$.** DBD employs a DB-score threshold $\tau$ to distinguish high-score examples (likely adversarial) from low-score examples (likely clean). We evaluate the effect of $\tau$ under three attack settings across 15 datasets, reporting the overall mean accuracy obtained by averaging both clean and adversarial performance across all datasets and attack settings (Fig.3(b)). The results show that setting $\tau = 0.8$ yields the best overall performance. At this threshold, DBD successfully reconstructs robust features for the majority of adversarial images, while simultaneously avoiding inaccurate shifts in most clean examples, thereby achieving a well-balanced trade-off between robust and clean accuracy. Additional results of the detection performance based on the DB-score threshold $\tau$ are shown in Fig.5 of the Appendix.

Table 6: **Ablation study.** Results (%) of clean accuracy (Acc.) and robust accuracy (Rob.) are average over 15 datasets and three attack settings. The left five columns correspond to different image transformation types, while the middle two columns represent DBD mechanisms (linear feature shifting and DB-score-based thresholding).

| Random Crop-Resize-Flip | Bit-depth Reduction | JPEG Compression | Guassian Noise | Image Filtering | Feature Shift | DB-score Threshold | Acc. | Rob. |
|---|---|---|---|---|---|---|---|---|
| - | - | - | - | - | - | - | 60.6 | 0.0 |
| ✓ | - | - | - | - | ✓ | - | 53.0 | 91.2 |
| - | ✓ | - | - | - | ✓ | - | 24.1 | 82.4 |
| - | - | ✓ | - | - | ✓ | - | 55.7 | 88.4 |
| - | - | - | ✓ | - | ✓ | - | 36.1 | 85.5 |
| - | - | - | - | ✓ | ✓ | - | 43.9 | 87.3 |
| ✓ | ✓ | ✓ | ✓ | ✓ | - | - | **61.5** | 34.8 |
| ✓ | ✓ | ✓ | ✓ | ✓ | ✓ | - | 58.9 | **92.1** |
| ✓ | ✓ | ✓ | ✓ | ✓ | ✓ | ✓ | 61.2 | 91.7 |

**Ablation of DBD mechanisms.** We conduct ablation experiments on all DBD components, with results average over 15 datasets and three attack settings (Table 6). Our analysis reveals four key findings: (1) Using a single type of image transformation with feature shifting provides strong adversarial defense but reduces clean image performance. (2) Aggregating multiple transformations without linear shifting gives the best clean accuracy but weak defense. (3) Adding linear shifting on top of multiple transformations substantially improves adversarial robustness, at the cost of some clean performance. (4) Applying the DB-score-based threshold to handle high-score and low-score examples separately achieves the best trade-off between clean accuracy and adversarial robustness.

# 5 CONCLUSION

In this work, we found that adversarial examples exhibit a strong directional bias under multiple input transformations, in contrast to the dispersed behavior of clean examples. Building on this observation, we proposed Directional Bias-guided Defense (DBD), a test-time framework that leverages the directional bias to reconstruct robust features through Defense Direction estimation and a two-stream reconstruction strategy based on the proposed DB-score. Experiments on 15 datasets under three attack settings demonstrate that DBD achieves state-of-the-art adversarial robustness while preserving zero-shot performance on clean images. Remarkably, robust accuracy even surpasses clean accuracy, highlighting that adversarial perturbations implicitly encode directional priors about the true decision boundary. We believe that our work sheds light on new perspectives for training-free defenses in VLMs.

## ACKNOWLEDGMENTS

This work was supported by the National Defense Technology Basic Scientific Research Project (Grant No. JSZL2023416A001).

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

# A APPENDIX

## A.1 ALGORITHM

We provide a detailed pseudocode of our proposed DBD in Algorithm 1.

---

**Algorithm 1:** Directional Bias-guided Defense (DBD) for VLMs

---

**Input:** Input image $x$, pre-trained CLIP vision encoder $\mathcal{E}_v$, classification head $\mathcal{H}$, transformation library $\mathcal{T}$, DB threshold $\tau$, shift factor $\lambda$

**Output:** Robust feature $\hat{f}$ and classification result $y$

**Step 1: Image Transformations & Feature Filtering**

**for** *transformation* $t \in \mathcal{T}$ **do**

$\quad$ $x_t \leftarrow t(x)$ ;                    // Apply transformation

$\quad$ $f_t \leftarrow \mathcal{E}_v(x_t)$ ;                    // Extract feature

$\quad$ $p_t \leftarrow \mathcal{H}(f_t)$ ;           // Compute classification probabilities

$\quad$ $E_t \leftarrow -\sum_c p_{t,c} \log p_{t,c}$ ;                    // Compute entropy

$\mathcal{F}_{\text{ref}} \leftarrow k$ features with lowest $E_t$

**Step 2: Directional Bias Computation**

$f_0 \leftarrow \mathcal{E}_v(x)$ ;                    // Original feature

**for** $f_i \in \mathcal{F}_{\text{ref}}$ **do**

$\quad$ $d_i \leftarrow (f_i - f_0)/\|f_i - f_0\|_2$ ;           // Unit displacement vector

$\bar{d} \leftarrow \frac{1}{k}\sum_{i=1}^k (f_i - f_0)$ ;                    // Mean displacement

$d_{\text{def}} \leftarrow \bar{d}/\|\bar{d}\|_2$ ;                    // Defense Direction

$S_{\text{db}} \leftarrow \frac{1}{k}\sum_{i=1}^k \langle d_i, d_{\text{def}}\rangle$ ;                    // DB-score

**Step 3: Two-stream Feature Reconstruction & Prediction**

**if** $S_{\text{db}} \leq \tau$ **then**

$\quad$ $\hat{f} \leftarrow f_0 + \|\bar{d}\|_2 \cdot d_{\text{def}}$ ;                    // Likely clean

**else**

$\quad$ $\hat{f} \leftarrow f_0 + \lambda \cdot \|\bar{d}\|_2 \cdot d_{\text{def}}$ ;        // Likely adversarial, enhanced shift

$y \leftarrow \text{argmax}(\mathcal{H}(\hat{f}))$ ;                    // Compute final prediction

---

## A.2 IMPLEMENTATION DETAILS

**Transformations.** In our experiments, DBD applies $n = 31$ transformations per input, yielding 32 images including the original. We provide more details on various image transformations used in DBD in Table 7. (1) For *spatial domain transformations*, each transformation includes random cropping, scaling, and flipping. Due to the stochastic nature of these operations, we perform 16 transformations per input. (2) For *pixel domain transformations*, we implement bit-depth reduction, compressing the original 8-bit image to 3 bits using three quantization methods: `floor`, `round`, and `ceil`. In addition, we apply JPEG compression-decompression with three quality levels (50, 60, 75) to increase diversity. Random Gaussian noise with amplitude $\gamma = 0.1$ is also added, with six stochastic realizations per input. (3) For *frequency domain transformations*, we apply Gaussian, mean, and median filtering with kernel size 5.

In the ablation study (Table 6 and Fig. 4), the resulting average features from Random Crop-Resize-Flip (16 transformations), Bit-depth Reduction (3 operations), JPEG Compression (3 operations), Gaussian Noise (6 operations), and Image Filtering (3 filters) are used for classification.

**Text Prompts.** For text prompts, we use a mix of hand-crafted prompts and GPT-3-generated prompts provided by CuPL (Pratt et al., 2023), and average text features over multiple prompts per class.

**Attacks.** All attacks are implemented using the torchattack (Kim, 2020) library. For PGD and FGSM attacks, we follow the baseline R-TPT (Sheng et al., 2025) configuration by using cross-entropy loss in untargeted mode. For the CW attack (Carlini & Wagner, 2017), we use a learning rate of 0.01 with the Adam optimizer. For AutoAttack (AA, (Croce & Hein, 2020b)), we use the standard mode, which includes four components: untargeted APGD-CE (1 restart), targeted APGD-DLR (10 target classes), targeted FAB (10 target classes, (Croce & Hein, 2020a)), and Square Attack (5000 queries, (Andriushchenko et al., 2020)). To remain consistent with the baseline setup, EOT was not applied. When verifying these 4 components, we keep the parameters unchanged.

Table 7: Details of image transformations used in DBD.

| Domain | Transformations |
|---|---|
| Spatial | Random cropping-scaling-flipping (16 times) |
| Pixel | Bit-depth compression (quantization): `floor, round, ceil` (bits = 3)
JPEG compression-decompression: `quality = 50, 60, 75`
Add Gaussian noise: $\gamma = 0.1$ (6 times) |
| Frequency | Gaussian filter: `kernel_size = 5`
Mean filter: `kernel_size = 5`
Median filter: `kernel_size = 5` |

## A.3 DATASETS

We evaluate our method on 10 fine-grained classification datasets and 5 ImageNet-OOD datasets. Table 8 summarizes their detailed information, including their content, number of categories, number of images and corresponding hand-crafted prompt.

Table 8: Introduction of all datasets involved in experiments.

| Dataset | Description | # Classes | # Test | Hand-crafted Prompt |
|---|---|---|---|---|
| Caltech101 | Object images | 100 | 2,465 | a photo of a [CLASS] |
| Pets | Pet images | 37 | 3,669 | a photo of a [CLASS], a type of pet |
| Cars | Car images | 196 | 8,041 | a photo of a [CLASS] |
| Flower102 | Flower images | 102 | 2,463 | a photo of a [CLASS], a type of flower |
| Aircraft | Aircraft images | 100 | 3,333 | a photo of a [CLASS], a type of aircraft |
| DTD | Describable textures images | 47 | 1,692 | [CLASS] texture |
| EuroSAT | Sentinel-2 satellite images | 10 | 8,100 | a centered satellite photo of a [CLASS] |
| UCF101 | Human action images | 101 | 3,783 | a photo of a person doing [CLASS] |
| SUN397 | Scene recognition images | 397 | 19,850 | a photo of a [CLASS] |
| Food101 | Food images | 101 | 30,300 | a photo of a [CLASS], a type of food |
| ImageNet | Object and scene images | 1,000 | 50,000 | a photo of a [CLASS] |
| ImageNet-A | Adversarially filtered images | 200 | 7,500 | a photo of a [CLASS] |
| ImageNet-V2 | New test images | 1,000 | 10,000 | a photo of a [CLASS] |
| ImageNet-R | Rendered images | 200 | 30,000 | a photo of a [CLASS] |
| ImageNet-S | Sketch-style images | 1,000 | 50,889 | a photo of a [CLASS] |

## A.4 EXPERIMENTS

### A.4.1 DETAILED RESULTS

**Detailed results under PGD attack with pseudo-label.** Due to space constraints, Table 3 in the main text reports only a subset of the results. Here we provide the complete version in Table 9, which includes results on six fine-grained datasets and three attack settings.

**Detailed results under various attacks.** We evaluate DBD and baseline methods under additional adversarial attacks, including FGSM (Goodfellow et al., 2014), CW (Carlini & Wagner, 2017), and

Table 9: robust accuracy (%) under three PGD attack settings using **pseudo-labels** on six fine-grained datasets.

| Attack & Model | Method | Caltech101 | Pets | Flower102 | Aircraft | DTD | UCF101 | Avg. |
|---|---|---|---|---|---|---|---|---|
| Clean | | 89.1 | 85.0 | 65.9 | 19.6 | 48.5 | 59.7 | 61.3 |
| PGD-100 ($\epsilon = 1/255$) on CLIP-ResNet50 | CLIP | 7.5 | 5.6 | 6.4 | 1.6 | 8.2 | 6.3 | 5.9 |
| | R-TPT | 81.7 | 77.7 | 53.2 | 15.1 | 35.6 | 54.3 | 52.9 |
| | DBD | 88.8 | 82.4 | 64.4 | 16.9 | 47.5 | 59.0 | 59.8 |
| Clean | | 93.3 | 86.6 | 67.0 | 20.6 | 49.9 | 63.6 | 63.5 |
| PGD-100 ($\epsilon = 1/255$) on CLIP-ViT-B/32 | CLIP | 2.3 | 3.4 | 3.8 | 0.8 | 3.8 | 4.9 | 3.2 |
| | R-TPT | 79.9 | 62.7 | 43.1 | 13.9 | 32.9 | 49.9 | 47.1 |
| | DBD | 93.0 | 84.4 | 66.1 | 19.5 | 49.5 | 62.7 | 62.5 |
| Clean | | 94.2 | 90.3 | 73.0 | 27.2 | 53.2 | 67.0 | 67.5 |
| PGD-100 ($\epsilon = 1/255$) on CLIP-ViT-B/16 | CLIP | 1.7 | 3.1 | 2.7 | 1.5 | 4.2 | 4.5 | 2.9 |
| | R-TPT | 84.1 | 66.1 | 49.8 | 18.2 | 37.1 | 52.7 | 51.3 |
| | DBD | 94.1 | 88.6 | 72.3 | 26.3 | 53.1 | 66.5 | 66.8 |

Table 10: Results (%) of clean accuracy (Acc.) and robust accuracy (Rob.) of various attacks on six fine-grained classification datasets.

| Method | Caltech101 | | | Pets | | | Flower102 | | | Aircraft | | | DTD | | | UCF101 | | | Avg. | | |
|---|---|---|---|---|---|---|---|---|---|---|---|---|---|---|---|---|---|---|---|---|---|
| | CLIP | R-TPT | DBD | CLIP | R-TPT | DBD | CLIP | R-TPT | DBD | CLIP | R-TPT | DBD | CLIP | R-TPT | DBD | CLIP | R-TPT | DBD | CLIP | R-TPT | DBD |
| **Attacks on CLIP-ResNet50** | | | | | | | | | | | | | | | | | | | | | |
| FGSM | 49.6 | 79.2 | 94.2 | 11.1 | 72.1 | 91.1 | 6.2 | 49.3 | 80.8 | 0.3 | 12.1 | 38.2 | 16.6 | 33.3 | 72.0 | 13.9 | 48.0 | 77.2 | 16.3 | 49.0 | 75.6 |
| CW | 5.1 | 79.6 | 90.6 | 0.8 | 74.7 | 86.6 | 0.9 | 51.2 | 65.6 | 0.9 | 14.9 | 18.8 | 2.2 | 33.7 | 50.1 | 2.4 | 51.5 | 62.0 | 2.1 | 50.9 | 62.3 |
| AA | 0.9 | 81.0 | 91.4 | 0.0 | 76.8 | 86.0 | 0.0 | 53.3 | 67.6 | 0.0 | 15.2 | 21.1 | 0.4 | 35.2 | 52.2 | 0.0 | 53.6 | 64.6 | 0.2 | 52.5 | 63.8 |
| FAB | 33.8 | 81.8 | 90.5 | 1.9 | 78.7 | 86.3 | 1.0 | 54.8 | 64.5 | 0.0 | 15.2 | 18.7 | 10.2 | 36.3 | 49.6 | 8.3 | 54.4 | 62.2 | 9.2 | 53.5 | 62.0 |
| Square | 50.5 | 83.2 | 87.7 | 36.5 | 80.4 | 83.1 | 33.7 | 57.7 | 56.5 | 1.2 | 15.8 | 15.6 | 19.6 | 37.3 | 44.4 | 18.1 | 56.7 | 56.3 | 26.6 | 55.2 | 57.3 |
| APGD-CE | 2.7 | 81.0 | 91.4 | 0.0 | 76.8 | 85.9 | 0.0 | 53.2 | 67.6 | 0.0 | 15.2 | 21.0 | 0.7 | 35.0 | 52.3 | 0.0 | 53.5 | 64.6 | 0.6 | 52.5 | 63.8 |
| APGD-DLR | 1.2 | 82.1 | 91.3 | 0.0 | 79.0 | 87.3 | 0.0 | 55.3 | 66.4 | 0.0 | 15.4 | 19.0 | 0.2 | 35.8 | 51.7 | 0.1 | 54.4 | 64.1 | 0.3 | 53.6 | 63.3 |
| **Attacks on CLIP-ViT-B/32** | | | | | | | | | | | | | | | | | | | | | |
| FGSM | 47.2 | 75.1 | 87.9 | 7.1 | 49.8 | 88.0 | 3.7 | 37.1 | 69.1 | 0.2 | 8.4 | 36.0 | 12.9 | 27.6 | 51.0 | 9.7 | 40.7 | 61.0 | 13.5 | 39.8 | 65.5 |
| CW | 3.0 | 84.7 | 92.8 | 0.4 | 73.9 | 87.5 | 0.9 | 55.3 | 69.1 | 0.2 | 16.8 | 22.9 | 1.5 | 36.4 | 50.7 | 1.4 | 55.9 | 64.8 | 1.2 | 53.8 | 64.6 |
| AA | 0.4 | 75.3 | 93.1 | 0.0 | 51.9 | 87.7 | 0.0 | 39.2 | 69.9 | 0.0 | 14.0 | 24.3 | 0.0 | 32.0 | 52.8 | 0.0 | 46.7 | 67.1 | 0.1 | 43.2 | 65.8 |
| FAB | 46.5 | 87.8 | 93.5 | 17.9 | 78.8 | 88.1 | 17.5 | 58.8 | 69.3 | 0.8 | 17.4 | 22.4 | 14.3 | 39.3 | 52.2 | 16.3 | 58.9 | 66.0 | 18.9 | 56.9 | 65.3 |
| Square | 22.7 | 86.7 | 92.2 | 8.0 | 77.6 | 84.7 | 8.9 | 57.9 | 65.6 | 0.5 | 17.0 | 20.4 | 3.8 | 37.4 | 48.3 | 6.9 | 58.9 | 64.0 | 8.4 | 55.9 | 62.5 |
| APGD-CE | 1.1 | 75.3 | 93.1 | 0.0 | 51.9 | 87.7 | 0.0 | 39.2 | 69.9 | 0.0 | 14.0 | 24.3 | 0.0 | 32.0 | 52.8 | 0.0 | 46.7 | 67.1 | 0.2 | 43.2 | 65.8 |
| APGD-DLR | 0.6 | 79.6 | 92.3 | 0.0 | 60.0 | 87.1 | 0.0 | 46.4 | 68.8 | 0.0 | 14.8 | 22.7 | 0.0 | 33.6 | 51.5 | 0.0 | 50.3 | 64.8 | 0.1 | 47.4 | 64.5 |
| **Attacks on CLIP-ViT-B/16** | | | | | | | | | | | | | | | | | | | | | |
| FGSM | 47.2 | 75.6 | 92.8 | 7.1 | 47.9 | 95.3 | 3.7 | 35.3 | 80.5 | 0.2 | 10.1 | 53.5 | 12.9 | 29.4 | 61.5 | 9.7 | 35.1 | 67.8 | 13.5 | 38.9 | 75.2 |
| CW | 3.0 | 87.3 | 94.7 | 0.4 | 73.4 | 92.1 | 0.9 | 55.7 | 73.9 | 0.2 | 20.0 | 28.8 | 1.5 | 39.5 | 56.2 | 1.4 | 57.4 | 69.2 | 1.2 | 55.6 | 69.1 |
| AA | 0.4 | 78.9 | 94.8 | 0.0 | 51.2 | 90.4 | 0.0 | 40.6 | 74.7 | 0.0 | 17.3 | 31.7 | 0.0 | 33.7 | 57.0 | 0.0 | 48.2 | 70.3 | 0.1 | 45.0 | 69.8 |
| FAB | 46.5 | 90.1 | 95.2 | 17.9 | 80.3 | 92.1 | 17.5 | 60.7 | 74.1 | 0.8 | 21.0 | 28.5 | 14.3 | 43.0 | 56.0 | 16.3 | 62.1 | 69.8 | 18.9 | 59.5 | 69.3 |
| Square | 22.7 | 89.6 | 93.5 | 8.0 | 77.7 | 87.8 | 8.9 | 62.9 | 68.8 | 0.5 | 21.2 | 24.7 | 3.8 | 41.3 | 50.4 | 6.9 | 62.5 | 65.3 | 8.4 | 59.2 | 65.1 |
| APGD-CE | 1.1 | 78.9 | 94.8 | 0.0 | 51.2 | 90.4 | 0.0 | 40.6 | 74.7 | 0.0 | 17.3 | 31.7 | 0.0 | 33.7 | 57.0 | 0.0 | 48.2 | 70.3 | 0.2 | 45.0 | 69.8 |
| APGD-DLR | 0.6 | 82.0 | 94.3 | 0.0 | 58.7 | 91.0 | 0.0 | 47.5 | 73.4 | 0.0 | 18.5 | 30.0 | 0.0 | 35.8 | 55.3 | 0.0 | 50.4 | 67.1 | 0.1 | 48.8 | 68.5 |

AutoAttack (AA) (Croce & Hein, 2020b). AA is a stronger, ensemble-based attack that combines targeted FAB (Croce & Hein, 2020a), Square Attack (Andriushchenko et al., 2020), untargeted APGD-CE (Auto-PGD with Cross-Entropy loss), and targeted APGD-DLR (Auto-PGD with Difference of Logits Ratio loss), incorporating both gradient-based and gradient-free strategies to comprehensively challenge model robustness. We also separately evaluate defense performance against each of these four constituent attacks. For our DBD, we set $\tau = 0.6$ and $\lambda = 2.5$ (with $\lambda = 5$ for FGSM). We assess the effectiveness of various attacks on six fine-grained datasets across three attack settings, and detailed results are shown in Table 10.

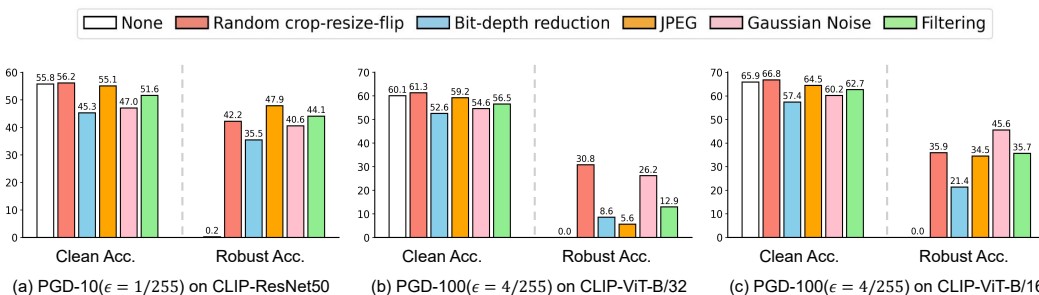

Figure 4: Average results (%) of clean accuracy (Clean Acc.) and robust accuracy (Robust Acc.) for various types of transformations across 15 datasets under three attack settings.

A.4.2 MORE ABLATION STUDIES

**Detailed analysis of various image transformations.** We ablate the individual image transformations used in DBD to assess their standalone effects (as Fig.4). JPEG compression-decompression preserves clean accuracy while providing moderate defense against low-strength attacks, though its effectiveness diminishes as attack strength increases. Random crop-resize-flip slightly improves clean accuracy and shows robustness across different attack strengths and model backbones. Adding Gaussian noise can yield strong defense in certain cases but substantially degrades clean accuracy. By combining multiple transformations, DBD leverages their complementary strengths to generate more reliable features across diverse scenarios and model variants; moreover, integration of transformations mitigates the risk of defenses being circumvented by attacks tailored to a single transformation.

**More ablation study on the DB-score threshold $\tau$.** We evaluate the detection performance of adversarial examples using the DB-score. Specifically, we assess the effect of threshold $\tau$ under three PGD attack settings across 15 datasets, reporting both the mean detection accuracy and mean F1-score averaged over all datasets and attack settings. Results are presented in Fig.5. The results show that $\tau = 0.8$ achieves near-optimal detection performance, and the metric remains stable in its neighborhood. This supports our choice and confirms that the DB-score provides a reliable signal for distinguishing adversarial from clean inputs.

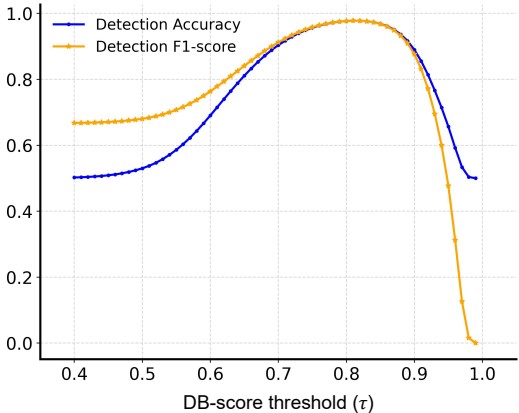

Figure 5: Average results of detection accuracy and F1-score for various types of transformations across 15 datasets under three attack settings.

**Adaptive Attack with BPDA and EOT.** While our primary evaluation based on the threat model described in Section 3.1, we additionally evaluate against adaptive, defense-aware attacks to assess potential vulnerabilities. Based on PGD attack, we implement BPDA (Backward Pass Differentiable Approximation, Athalye et al. (2018)) combined with EOT (Expectation Over Transformation, (Xie et al., 2017)) to approximate gradients through our differentiable DBD pipeline. During attack optimization, non-differentiable components are replaced with identity functions in the backward pass, while gradients are averaged over multiple stochastic forward passes.

We evaluate on Caltech101 dataset using strong adaptive attacks: PGD-10 ($\epsilon$=1/255) against CLIP-ResNet50 and PGD-100 ($\epsilon$=4/255) against CLIP-ViT-B/16. Results in Table 11 show that when attackers explicitly optimize through the full DBD pipeline, robust accuracy degrades significantly (to 50.79% and 1.29%, respectively). Notably, these adversarial images generated buy adaptive attacks also perform substantially worse against the original CLIP model compared to standard attacks, suggesting overfitting to the defense mechanism.

Critically, under the adaptive attack (PGD-100), the average DB-score remains high (0.89), with approximately 81% of adversarial samples exceeding our detection threshold ($\tau$=0.80). This reveals an important insight: *the directional bias pattern persists under adaptive attacks, but the estimated direction is manipulated to point away from the true class*. In other words, while adaptive attacks

can subvert the defense functionality by distorting the Defense Direction, they cannot eliminate the underlying directional bias signal, making such attacks still detectable through our DB-score metric.

Table 11: Robust accuracy (%) under adaptive attacks with BPDA+EOT on Caltech101.

| Attacks | Original CLIP | CLIP with DBD |
|---|---|---|
| PGD-10 ($\epsilon$=1/255) | 84.26 | 50.79 |
| PGD-100 ($\epsilon$=4/255) | 81.05 | 1.29 |

**Analysis of attacks using different loss functions.** To evaluate the robustness of our method in more practical and challenging settings where ground-truth labels are unavailable during inference, we consider margin-based adversarial attacks, which does not rely on true class labels but instead maximizes the difference between the highest non-target logit and the target logit. We conduct PGD-100 attacks ($\epsilon = 4/255$) using both cross-entropy (CE) loss and margin loss, with either ground-truth labels or model-generated pseudo-labels, on the Caltech101 dataset. Results are shown in Table 12. Notably, DBD achieves 99.76% robust accuracy under margin-loss attacks with ground-truth guidance, on par with its performance under CE-loss attacks (99.39%). More importantly, when using pseudo-labels (i.e., no access to ground truth at test time), DBD maintains strong robustness (94.24% vs. 94.12% under CE-loss), demonstrating that the directional bias exploited by our method is not specific to a particular attack loss formulation. In contrast, the vanilla CLIP model collapses under all attack variants (<3% robust accuracy). These results confirm that DBD's defense mechanism generalizes across different adversarial objectives and remains effective in realistic label-free scenarios.

Table 12: Robust accuracy (%) under PGD-100 attacks ($\epsilon = 4/255$) on Caltech101 using different loss functions and label sources.

| Attack Setting | CLIP | DBD (ours) |
|---|---|---|
| CE-loss + Ground-truth | 0.04 | 99.39 |
| CE-loss + Pseudo-label | 1.66 | 94.12 |
| Margin-loss + Ground-truth | 0.04 | 99.76 |
| Margin-loss + Pseudo-label | 2.80 | 94.24 |

**Analysis of Correct vs. Incorrect Clean Predictions.** We separate samples into those correctly classified by CLIP on clean images versus those misclassified, and evaluate robust accuracy under PGD-100 attacks ($\epsilon = 4/255$) on CLIP-ViT-B/16. Results are shown in Table 13. The near-perfect robust accuracy on initially correct samples demonstrates that DBD effectively preserves accurate predictions by leveraging directional information from adversarial perturbations. More importantly, the high robust accuracy on initially misclassified samples (up to 95.83%) shows that DBD genuinely *corrects* errors by exploiting the directional prior embedded in the attack. This dual capability explains why our overall robust accuracy exceeds clean accuracy, revealing an important insight: ground-truth guided attacks inadvertently leak information about the correct class direction.

Table 13: Robust accuracy (%) under PGD-100 attacks on correctly and incorrectly predicted clean samples.

| | Caltech101 | Pets | Flower102 | DTD | UCF101 |
|---|---|---|---|---|---|
| Correct clean predictions | 99.61% | 97.83% | 99.11% | 99.00% | 98.74% |
| Incorrect clean predictions | 95.83% | 90.14% | 93.08% | 84.60% | 90.14% |

**Analysis of inference efficiency.** We compare DBD's inference efficiency with baseline methods. Table 14 reports the training time of APT (Li et al., 2024) and the inference time of test-time defenses R-TPT (Sheng et al., 2025), TTC (Xing et al., 2025), and DBD on UCF101 using CLIP-ViT-B/32. Training-time defenses incur substantial training costs, while test-time defenses avoid this but increase inference overhead. DBD achieves a favorable balance between robustness and efficiency, outperforming R-TPT in both accuracy and inference speed.

Table 14: Running time and robust accuracy (%) of different defense methods against adversarial attacks on UCF101 dataset using CLIP-ViT-B/32. APT is evaluated with 16 shots, while R-TPT and DBD are evaluated with 32 views.

| Method | Stage | Running time | Rob. |
|--------|-------|--------------|------|
| APT | Training time | 22m47s / 200 epochs | 18.9 |
| TTC | Test time | 0.008s / image | 27.6 |
| R-TPT | Test time | 0.181s / image | 44.3 |
| DBD | Test time | 0.025s / image | **92.2** |

USE OF LLMS

In this work, we used ChatGPT to assist in polishing the writing of this paper, focusing primarily on improving clarity, grammar, and style. The model was not involved in the generation of ideas or experimental designs. All the concepts, analyses, and conclusions presented are entirely our own.

