# OpenReview forum: "Adversarial Attacks Already Tell the Answer: Directional Bias-Guided Test-time Defense for Vision-Language Models"
_ICLR.cc/2026/Conference — ICLR 2026 Poster_

### Official Review · Reviewer_H8wX · 2025-10-18

**Soundness:** 2
**Presentation:** 2
**Contribution:** 2
**Rating:** 4
**Confidence:** 4

**Summary:**

This paper introduces an interesting observation that adversarial features exhibit a "directional bias" under input transformations. Based on this, it proposes DBD, a simple and training-free test-time defense that estimates and reverses this bias. The method achieves state-of-the-art robustness, with reported robust accuracy **even surpassing** clean accuracy under certain conditions.

**Strengths:**

- The core observation of a "Defense Direction" for adversarial examples is novel and insightful.

- The proposed method is simple, intuitive, and computationally efficient as a test-time defense.

**Weaknesses:**

- Unrealistic threat model: The paper's most impressive results are predicated on a PGD attack that requires access to ground-truth labels. This is a well-known weak and impractical threat model that leaks information about the correct class, which the defense is perfectly tailored to exploit. The validity of the core claims hinges on this flawed setup. In other words, the white box + cross-entropy loss configuration was originally established within the community for stress testing (previously the strongest achievable attack effect). If a model performs well under this configuration, there is reason to expect it will also perform well under other configurations. However, this method alters that situation by meticulously designing the approach specifically for this stress-testing scenario, rather than for more general scenarios.

- The headline claim that "robust accuracy can surpass clean accuracy" is an artifact of the unrealistic threat model. The more realistic pseudo-label evaluation, where this phenomenon vanishes and performance drops significantly, is not presented as the main result, which is misleading.

- The "directional bias" hypothesis is well-motivated for gradient-based attacks. However, the paper lacks a convincing analysis of why this mechanism is also effective against AutoAttack, potentially weakening the claimed technical contribution.

**Questions:**

- Could the authors justify centering the main evaluation on a threat model that requires access to ground-truth labels? To what extent does the high robustness achieved in this setting transfer to more realistic attack scenarios where such information is unavailable to the adversary?

- If the results from the more realistic pseudo-label attack evaluation (Table 3) were presented as the paper's primary findings, how would the authors reframe the central claims and contributions of this work?

- The "Defense Direction" hypothesis is well-motivated for speicific gradient-based attacks. Could you explain why this is also effective against the gradient-free components of AutoAttack, beyond a general robustness improvement from ensembling multiple input transformations?

---

> ### Author Response · Authors · 2025-11-20
> **Official Comment by Authors (Part 1)**
>
> We thank the insightful review and would like to address the issue by points:
>
> > W1 and Q1:  Could the authors justify centering the main evaluation on a threat model that requires access to ground-truth labels? To what extent does the high robustness achieved in this setting transfer to more realistic attack scenarios where such information is unavailable to the adversary?
>
> Thank you for this insightful question. We agree that evaluating solely under a ground-truth threat model has limitations, and we appreciate the opportunity to clarify our position.
>
> **Regarding evaluation protocol**: As noted, PGD with ground-truth labels has become the widely adopted standard in test-time defense literature (including SOTA methods like R-TPT). This setting enables strong, targeted attacks that can reduce vanilla CLIP’s robust accuracy to near zero—providing a clear benchmark for measuring defense strength. For fair comparison with prior work, we followed this convention.
>
> **Regarding realism **: We also consider this threat model unrealistic, as it grants attackers excessively strong prior knowledge (ground-truth labels), creating a risk of information leakage. Our study reveals a counter-intuitive finding: our defense can actually *leverage this prior information* to achieve robust accuracy exceeding clean accuracy. This result serves as an important reminder to the community: overly strong attack assumptions may inadvertently create detectable patterns that sophisticated defenses can exploit.
>
> More importantly, **our core insight—that *adversarial attacks inherently encode directional priors about the true decision boundary*—holds across threat models**. Consider:
>
> - With **ground-truth labels**, attacks contain near-perfect directional prior (100% correct), enabling DBD to achieve >90% robust accuracy.
> - With **pseudo-labels** (more realistic), only correctly classified clean samples provide valid directional priors (proportional to clean accuracy). As Table 3, DBD still recovers robust accuracy (66.8%) close to clean accuracy (67.5%).
>
>
>
> > W2 and Q2:  If the results from the more realistic pseudo-label attack evaluation (Table 3) were presented as the paper's primary findings, how would the authors reframe the central claims and contributions of this work?
>
> Thank you for this important question. **To clarify, our central claim and contribution is: we propose the core insight that adversarial attacks inherently encode directional priors about the true decision boundary, and build the DBD pipeline to exploit this geometric property for test-time defense.** Crucially, as demonstrated in both ground-truth and pseudo-label settings, this insight—and our method’s effectiveness—holds across threat models.
>
> Regarding "robust accuracy can surpass clean accuracy": this phenomenon emerges exclusively under ground-truth label attacks—a setting widely adopted in existing literature. We position this result not as our core contribution, but as a surprising experimental finding based on our fundamental insight and proposed DBD method. This surprising finding serves as both validation of our approach and a warning to the community.

---

> > ### Author Response · Authors · 2025-11-20
> > **Official Comment by Authors (Part 2)**
> >
> > > W3 and Q3: The "Defense Direction" hypothesis is well-motivated for speicific gradient-based attacks. Could you explain why this is also effective against the gradient-free components of AutoAttack, beyond a general robustness improvement from ensembling multiple input transformations?
> >
> > Thank you for this insightful comment. We clarify that our Defense Direction is not derived from gradient alignment with specific attacks, but from a more fundamental observation: for an adversarial attack to succeed, it must push the image feature out of the correct decision region. This displacement inherently encodes directional information pointing back toward the correct class.
> >
> > Our method leverages diverse image transformations to suppress adversarial noise and estimate this directional bias. As such, the bias itself is attack-agnostic. However, under common and efficient gradient-based PGD attacks, the large shift in image features makes the directional bias more pronounced. In contrast, AutoAttack combines multiple components—including APGD with adaptive step sizes and black-box attacks—which results in smaller feature shifts for some inputs. This makes the directional bias less consistent and the estimated Defense Direction less accurate. Although the improvement is not as pronounced as against PGD, our method remains highly effective against AutoAttack .
> >
> > To causally validate the geometric validity of the estimated Defense Direction, we compute its cosine similarity with two reference directions:
> >
> > - **Clean Direction**: defined as the direction from the adversarial image’s feature to the feature of its corresponding clean (unperturbed) counterpart.
> > - **Class Centroid Direction**: defined as the direction from the adversarial image’s feature to the true class centroid of features, where the centroid is computed using correctly classified clean images.
> >
> > Results on Pets Dataset are show below:
> >
> > |                                                      | PGD    | AutoAttack |
> > | ---------------------------------------------------- | ------ | ---------- |
> > | Clean Direction (average  cosine similarity)         | 0.957  | 0.825      |
> > | Class Centriod Direction (average cosine similarity) | 0.932  | 0.829      |
> > | DB-score (average)                                   | 0.966  | 0.904      |
> > | Robust accuracy                                      | 97.08% | 89.75%     |
> >
> > These high cosine similarities provide strong geometric evidence that the Defense Direction indeed aligns closely with both the clean feature direction and the correct class centroid—supporting our hypothesis that Defense Direction points back toward the correct decision region.

---

> > ### Comment · Reviewer_H8wX · 2025-11-27
> >
> > I generally agree with the authors’ position: if this work is understood as a mechanism-discovery study, the contribution is meaningful; however, framing it as a stronger defense seems not enought. To better assess the claimed contribution, I would like the authors to explicitly clarify several scenarios that I am not sure were fully covered in the current manuscript:
> >
> >  **Attacker with 100% access to ground-truth labels.**
> > I believe this corresponds to the main experimental setting, but I would appreciate a concise summary of what happens in this idealized case and how the proposed method behaves relative to standard supervision-driven attacks.
> >
> > **Pseudo-label scenarios.**
> > Suppose the attacker relies on unsupervised or weakly supervised clustering to obtain pseudo-labels, potentially with mismatched label cardinality (e.g., true 10-class dataset → clustering produces 100 clusters). How would such label inconsistency affect the attack process and the performance of the proposed defense mechanism?
> >
> > **No-attack (clean input) scenario  BUT under the defense assumption**.
> > **This is the case I am most interested in.** The current defense implicitly assumes that “the attacker generates adversarial examples using certain labels.” What happens if the input image is not adversarial at all, yet the defender still applies the full defense pipeline under this assumption?
> > Does the method introduce systematic distortions?
> > Is there a risk of degrading clean accuracy?
> > How robust is the method to such false-positive defense triggering?
> > Clarifying the above settings, especially the third one, would significantly improve the paper’s completeness and help readers understand the practical applicability and limitations of the proposed approach.

---

> > > ### Author Response · Authors · 2025-11-29
> > > **Response by Authors**
> > >
> > > We sincerely appreciate your thoughtful follow-up review and your recognition of the value of our mechanism-discovery perspective. Below, we would like to address each of your remaining concerns with further analysis and clarification.
> > >
> > > > Attacker with 100% access to ground-truth labels:
> > >
> > > In this standard setting, attackers can reduce model accuracy to near 0% using methods like PGD or AutoAttack. Our defense effectively leverages the strong directional prior embedded in these powerful attacks, achieving high robust accuracy. Notably, under PGD attacks, our method sometimes achieves higher robust accuracy than the clean accuracy baseline—a phenomenon demonstrating how effectively we harness the directional information contained in adversarial perturbations.
> > >
> > > > Pseudo-label scenarios.
> > >
> > > Our current experiments used CLIP's predictions as pseudo-labels within a fixed class set. We acknowledge that clustering-based pseudo-label generation with mismatched cardinality (e.g., 10 true classes vs. 100 clusters) represents an interesting scenario not fully explored in our work. When attackers and defenders operate with significantly different class cardinalities, the resulting adversarial examples typically transfer poorly to the original CLIP model, weakening the attack and limiting their practical impact. Since our method relies on strong attacks that create pronounced directional bias, weaker attacks from label mismatch would reduce our Defense Direction's estimation accuracy. This would be a valuable direction for future work.
> > >
> > > > No-attack (clean input) scenario BUT under the defense assumption. This is the case I am most interested in.
> > >
> > > Our method incorporates a DB-score threshold to distinguish likely adversarial samples (high DB-score) from clean samples (low DB-score), preventing unnecessary feature reconstruction on clean inputs. **As shown in Table 6, our method maintains clean accuracy almost unchanged (61.5% → 61.2%) while dramatically improving robust accuracy (0% → 91.7%).**
> > >
> > > Additionally, our reconstruction process uses a relative shift distance based on the displacement from original feature to the mean of transformed features, which naturally limits distortion on clean samples. **Even when forcing reconstruction on all clean images (bypassing the DB-score threshold), accuracy only decreases modestly (61.5% → 58.9%)**, demonstrating our method's resilience to false-positive defense triggering.

---

### Official Review · Reviewer_fqTJ · 2025-10-20

**Soundness:** 3
**Presentation:** 3
**Contribution:** 2
**Rating:** 4
**Confidence:** 4

**Summary:**

The paper identifies a dominant feature-space direction for adversarial images under diverse transformations, quantified via a DB-score, and proposes DBD to reconstruct features by shifting along a “Defense Direction” (high-DB) or averaging (low-DB). Evaluations on 15 datasets and multiple attacks (including AA on a subset) report high robust accuracies, sometimes exceeding clean accuracy. The approach is training-free, simple, and efficient relative to some prompt-tuning defenses.

**Strengths:**

- Clear empirical phenomenon and DB-score metric with intuitive visualizations
- Simple, training-free mechanism with precise equations
- Inclusion of all CLIP backbone tests and AutoAttack, as well as efficiency comparisons
- $\lambda$/$\tau$ ablations are helpful
- most hyperparameters are included for reproducibility, but the exact AA configs, seeds, and adaptive scripts are not provided even in the appendix, thus reproducibility concerns remain

**Weaknesses:**

- No adaptive white-box evaluation (BPDA/EOT) against the full defense despite stochastic and non-smooth components
- AutoAttack scope is limited and apparently non-adaptive to DBD; targeted AA not analyzed
- Robust > clean claims require stronger calibration evidence to rule out masking
- Causal validation of “Defense Direction” via gradient alignment is missing
- Timing numbers seem optimistic for 31 transforms; measurement details needed
- Prompt-set sensitivity not ablated

**Questions:**

- Can you add BPDA+EOT attacks with AA, differentiating through expectation over transforms, entropy filtering, and reconstruction, plus ε/step monotonicity curves?
- Which specific AA modes and losses were used? was EOT applied? any targeted AA results?
- Can you provide resource costs including (batching, precision, forward counts) and FLOPs?
- Can you ablate prompt sets and template counts (hand-crafted vs CuPL) and report multi-seed confidence intervals?

---

> ### Author Response · Authors · 2025-11-20
> **Official Comment by Authors (Part 1)**
>
> We thank the insightful review and would like to address the issue by points:
>
> > W1 and Q1: No adaptive white-box evaluation (BPDA/EOT) against the full defense despite stochastic and non-smooth components. Can you add BPDA+EOT attacks with AA, differentiating through expectation over transforms, entropy filtering, and reconstruction, plus ε/step monotonicity curves?
>
> Thank you for this important suggestion. We apologize for any possible misunderstanding, but we would like to clarify that our experimental setup follows the evaluation protocol of the recent SOTA test-time defense R-TPT (CVPR 2025) [1] for VLMs. Specifically, we assume the attacker has full access to the CLIP model weights and crafts adversarial examples against the *original* CLIP model, without knowledge of the test-time defense mechanism or its parameters. This setting reflects a realistic scenario: foundation models like CLIP are widely used in downstream applications and their weights are publicly available, whereas test-time defenses are typically deployed privately and are not exposed to attackers. We have revised the relevant descriptions in Section 3.1 to reduce potential misunderstandings.
>
> We agree that adaptive or defense-aware attacks are an important consideration. *Due to time constraints, we were unable to implement the more complex AutoAttack with BPDA [2] and EOT [3] experiments. As an alternative, we have conducted simplified experiments using PGD combined with BPDA and EOT, which we hope adequately addresses your concerns.* Specifically, during attack optimization, we treat the entire DBD pipeline as part of the defended model. Since some image transformations in DBD are non-differentiable, we use the identity function in the backward pass (BPDA) and average gradients over multiple stochastic forward passes (EOT) to approximate the true gradient.
>
> We evaluated on Caltech101 dataset using two settings:
>
> - PGD-10 (*ϵ*=1/255) against CLIP-ResNet50
> - PGD-100 (*ϵ*=4/255) against CLIP-ViT-B/16
>
> The results of robust accuracies (%) are as follows:
>
>
> |                           | Original CLIP | CLIP with DBD |
> | ------------------------- | ------------- | ------------- |
> | PGD-10($\epsilon=1/255$)  | 84.26         | 50.79         |
> | PGD-100($\epsilon=4/255$) | 81.05         | 1.29          |
>
> As expected, when the attacker explicitly optimizes through the full DBD pipeline, our defense can be significantly degraded. However, adversarial examples generated by these adaptive attacks achieve significantly lower attack success rates against the original undefended CLIP model compared to standard attacks. This suggests the attack is overfitting to the DBD mechanism and loses transferability to the base classifier.
>
> **Moreover, under the adaptive attack (PGD-100, $\epsilon$=4/255), the average DB-score remains high (0.89), and approximately 81% of adversarial samples have a DB-score > threshold (0.80). This indicates that the *directional bias pattern persists*, but the estimated direction no longer aligns with the true class.** In other words, the attacker successfully manipulates the bias to point elsewhere, rendering it ineffective as a Defense Direction, though still detectable. We have incorporated this experiment into the appendix of the revised paper.
>
> > W2 and Q2: AutoAttack scope is limited and apparently non-adaptive to DBD; targeted AA not analyzed. Which specific AA modes and losses were used? was EOT applied? any targeted AA results?
>
> We apologize for omitting these details in the original paper, and have supplemented this content in the appendix (Section A.2) of the revised paper. For AutoAttack, we use the standard mode, which includes four components: **untargeted APGD-CE** (1 restart), **targeted APGD-DLR** (10 target classes), **targeted FAB** (10 target classes), and **Square Attack** (5000 queries). To remain consistent with the baseline setup, EOT was not applied. The results of the AutoAttack and the four components are provided in Table 4, where our method achieves robust accuracy that is close to or exceeds clean accuracy, demonstrating the effectiveness of our approach.

---

> > ### Author Response · Authors · 2025-11-20
> > **Official Comment by Authors (Part 2)**
> >
> > > W3: Robust > clean claims require stronger calibration evidence to rule out masking
> >
> > Thank you for this insightful question. The reason why *robust accuracy can exceed clean accuracy* is that DBD not only preserves correct predictions on samples that the model classifies correctly under clean conditions, but also successfully recovers many samples that the model misclassifies even without any attack. For these hard samples, their clean features already lie in an incorrect decision region. However, when PGD generates adversarial examples using the **ground-truth label**, the perturbation still follows the gradient that moves features in the *opposite* direction—toward the true class. Our method captures this shift and reconstructs features along this direction, effectively pulling these samples back into the correct decision region. This leads to previously misclassified samples becoming correctly classified after defense, thereby producing robust accuracy higher than clean accuracy. This phenomenon also indicates that ground-truth–guided adversarial perturbations implicitly encode a useful directional prior about the true decision boundary, which DBD is able to exploit.
> >
> >
> >
> > > W4: Causal validation of “Defense Direction” via gradient alignment is missing
> >
> > Thank you for this insightful comment. We clarify that our Defense Direction is not derived from gradient alignment with specific attacks, but from a more fundamental observation: for an adversarial attack to succeed, it must push the image feature out of the correct decision region. This displacement inherently encodes directional information pointing back toward the correct class.
> >
> > Our method leverages diverse image transformations to suppress adversarial noise and estimate this directional bias. As such, the bias itself is attack-agnostic. However, under common and efficient gradient-based PGD attacks, the large shift in image features makes the directional bias more pronounced. In contrast, AA combines multiple components—including APGD with adaptive step sizes and black-box attacks—which results in smaller feature shifts for some inputs. This makes the directional bias less consistent and the estimated Defense Direction less accurate. Although the improvement is not as pronounced as against PGD, our method remains highly effective against AA.
> > To validate the geometric validity of the estimated Defense Direction, we compute its cosine similarity with two reference directions:
> >
> > - **Clean Direction**: defined as the direction from the adversarial image’s feature to the feature of its corresponding clean (unperturbed) counterpart.
> > - **Class Centroid Direction**: defined as the direction from the adversarial image’s feature to the true class centroid of features, where the centroid is computed using correctly classified clean images.
> >
> > Results on Pets Dataset are show below:
> >
> > |                                                      | PGD    | AA     |
> > | ---------------------------------------------------- | ------ | ------ |
> > | Clean Direction (average  cosine similarity)         | 0.957  | 0.825  |
> > | Class Centriod Direction (average cosine similarity) | 0.932  | 0.829  |
> > | DB-score (average)                                   | 0.966  | 0.904  |
> > | Robust accuracy                                      | 97.08% | 89.75% |
> >
> > These high cosine similarities provide strong geometric evidence that the Defense Direction indeed aligns closely with both the clean feature direction and the correct class centroid—supporting our hypothesis that Defense Direction points back toward the correct decision region.
> >
> >
> >
> > > W5: Timing numbers seem optimistic for 31 transforms; measurement details needed
> >
> > Thank you for raising this important practical concern. We used 31 transformations not because this number is optimal for deployment, but to reliably demonstrate the directional bias phenomenon and obtain a stable estimate of the Defense Direction across diverse attack types. As shown in **Appendix Figure 4**, no single type of transformation consistently suppresses adversarial noise across different attack settings—motivating our ensemble approach for experimental validation.
> >
> > **Table 6 (top six rows)** confirms that even a single transformation combined with DB shift yields non-trivial robustness (though lower than the full ensemble). For practical deployment, users can trade off speed and robustness by selecting a smaller subset of transformations based on their threat model.

---

> ### Author Response · Authors · 2025-11-20
> **Official Comment by Authors (Part 3)**
>
> > Q3: Can you provide resource costs including (batching, precision, forward counts) and FLOPs?
>
> Thank you for this practical question. As a test-time defense, our method operates with the following resource profile:
>
> - Batching: Inference is performed with batch size = 1 (per input image).
> - Transformations: Each input generates 32 transformed variants (including the original). These transformations are preprocessing operations executed on CPU and can be parallelized via multi-threading with minimal overhead.
> - Precision: CLIP features are extracted in FP32 precision.
> - Forward passes: Each sample requires exactly one forward pass through CLIP for each transformed image (32 total), with no backward passes or iterative optimization.
> - FLOPs: CLIP-ViT-B/16 requires 5.82 GFLOPs per (224×224×3) image. For 32 transformed images, the total is 186.24 GFLOPs per sample. The subsequent Defense Direction computation and feature reconstruction add negligible FLOPs (<0.1 GFLOPs).
>
> Runtime measurements (reported in Table 5) show an average inference latency of 0.025 seconds per image on UCF101 (RTX 3090 GPU). This demonstrates a favorable trade-off between robustness gains and computational cost for real-world deployment.
>
>
>
>
> > W6 and Q4: Prompt-set sensitivity not ablated. Can you ablate prompt sets and template counts (hand-crafted vs CuPL) and report multi-seed confidence intervals?
>
> Thank you for this valuable suggestion. We conducted a prompt-set ablation on the DTD dataset under PGD-100 (*ϵ*=4/255,*α*=1/255) with CLIP-ViT-B/16. Results are averaged over **3 random seeds** with 95% confidence intervals:
>
> |              | clean accuracy(%) | robust accuracy(%) |
> | ------------ | ----------------- | ------------------ |
> | hand-crafted | 45.11$\pm$0.85    | 79.70$\pm$0.44     |
> | CuPL [4]     | 54.45$\pm$0.96    | 92.29$\pm$0.07     |
>
> Key observations:
>
> 1. CuPL significantly boosts baseline performance (clean accuracy +9.34%), confirming its effectiveness for fine-grained tasks.
> 2. DBD maintains strong robustness under both prompt settings, with robust accuracy consistently exceeding clean accuracy.
> 3. The small confidence intervals demonstrate stable performance across seeds.
>
> [1] Lijun Sheng, Jian Liang, Zilei Wang, and Ran He. R-tpt: Improving adversarial robustness of vision-language models through test-time prompt tuning. In Proceedings of the Computer Vision and Pattern Recognition Conference, pp. 29958–29967, 2025.
>
> [2] Anish Athalye, Nicholas Carlini, and David Wagner. Obfuscated gradients give a false sense of security: Circumventing defenses to adversarial examples. In International conference on machine learning, pp. 274–283. PMLR, 2018.
>
> [3] Cihang Xie, Jianyu Wang, Zhishuai Zhang, Zhou Ren, and Alan Yuille. Mitigating adversarial effects through randomization. arXiv preprint arXiv:1711.01991, 2017.
>
> [4] Sarah Pratt, Ian Covert, Rosanne Liu, and Ali Farhadi. What does a platypus look like? generating customized prompts for zero-shot image classification. In Proceedings of the IEEE/CVF international conference on computer vision, pp. 15691–15701, 2023.

---

> > ### Comment · Reviewer_fqTJ · 2025-11-27
> >
> > Thank you for the very substantial effort in the rebuttal and for running the extra geometric and simplified BPDA+EOT experiments, which improve the empirical picture and help clarify the intended threat model. However, I still see several core issues that limit how strong the conclusions can be. First, the adaptive evaluation with **BPDA+EOT** on a single dataset shows that DBD can be driven to very low robust accuracy once the attacker explicitly optimizes through the full pipeline, which suggests that the proposed **directional prior** is fragile when the defense is fully known, even if such attacks are positioned as outside the primary threat model. Second, the central claim that adversarial gradients already point toward the correct class region in CLIP space remains mostly **intuitive** and empirical, without a theoretical account or a clear explanation of the underlying mechanism and of the conditions under which this behavior should be expected to hold. Third, the analysis does not separate cases where DBD simply preserves already correct clean predictions from cases where it actually corrects wrong clean predictions, and therefore does not clearly demonstrate that DBD is genuinely exploiting useful directional information from the attack. It also does not study **non label guided attacks** based on model predictions or margin losses, which would be more representative for deployed vision language systems that lack ground truth labels at test time, so the practical **generality** of the claimed directional prior remains uncertain. Finally, in the VLM experiments the prompt analysis compares only **hand crafted prompts** and **CuPL**, without including **learnable prompt** methods such as **CoOp** that explicitly optimize class embeddings and could already capture similar directional effects. Taken together, these points make the empirical phenomenon interesting, but the defense and its associated **robustness** claims still feel too narrow and fragile in the current version.

---

> > > ### Author Response · Authors · 2025-11-29
> > > **Response by Authors (Part1)**
> > >
> > > We sincerely appreciate your thoughtful follow-up review and recognition of our additional experimental efforts. Below, we would like to address each of your remaining concerns with further analysis and clarification.
> > >
> > > > First, the adaptive evaluation with **BPDA+EOT** on a single dataset shows that DBD can be driven to very low robust accuracy once the attacker explicitly optimizes through the full pipeline, which suggests that the proposed **directional prior** is fragile when the defense is fully known, even if such attacks are positioned as outside the primary threat model.
> > >
> > > Thank you for your thoughtful critique regarding the adaptive evaluation with BPDA+EOT. We agree that the directional prior can be fragile when the defense is fully known to the attacker. However, we note that adversarial examples generated through these targeted attacks transfer poorly to the original undefended CLIP model, limiting their practical impact.
> > >
> > > Moreover, the table below shows the average DB-scores computed on clean images, standard PGD adversarial examples, and PGD examples with BPDA+EOT against our full pipeline:
> > >
> > > |               | Clean  |  PGD   | PGD(with BPDA+EOT) |
> > > | ------------- | :----: | :----: | :----------------: |
> > > | CLIP-ResNet50 | 0.6476 | 0.9072 |       0.7859       |
> > > | CLIP-ViT-B/16 | 0.6611 | 0.9616 |       0.8870       |
> > >
> > > Attack settings: PGD-10 ($\epsilon$=1/255) for CLIP-ResNet50; PGD-100 ($\epsilon$=4/255) for CLIP-ViT-B/16, on Caltech101 dataset.
> > >
> > > Even under adaptive attacks, the DB-score remains significantly higher than on clean images (0.7859 vs. 0.6476 for ResNet50; 0.8870 vs. 0.6611 for ViT-B/16). **This demonstrates that while adaptive attacks can compromise the accuracy of Defense Direction estimation, the underlying directional bias phenomenon persists.** Importantly, this distinctive signature remains detectable, allowing our method to identify such adaptive attacks through DB-score analysis despite their optimization against our full pipeline.
> > >
> > >
> > >
> > > > Second, the central claim that adversarial gradients already point toward the correct class region in CLIP space remains mostly **intuitive** and empirical, without a theoretical account or a clear explanation of the underlying mechanism and of the conditions under which this behavior should be expected to hold.
> > >
> > > Thank you for this insightful question about the theoretical foundation of our directional bias claim. **We clarify that our Defense Direction is not derived from adversarial gradient alignment with specific attacks, but stems from a fundamental geometric principle: for an adversarial attack to successfully fool a model on a previously correctly classified sample, it must push the image feature outside the correct decision region. This displacement inherently encodes directional information pointing back toward the true class.**
> > >
> > > This principle holds specifically when attacks target samples that were correctly classified by the original CLIP model. In such cases, the adversarial perturbation necessarily moves features away from the correct decision boundary, creating a detectable directional bias back toward the true class. However, for samples initially misclassified by CLIP, the feature already resides outside the correct decision region. Here, attacks behave differently: some attacks (eg. PGD) push features further away from the true class, while others (eg. C&W, AA) shows less consistent directional patterns as it prioritizes minimal perturbation over directional displacement.
> > >
> > > Our method leverages multiple image transformations to estimate the Defense Direction, and we provided geometric verification of its alignment with the correct class region in our previous response (Part2, W4).

---

> > > > ### Author Response · Authors · 2025-11-29
> > > > **Response by Authors (Part2)**
> > > >
> > > > > Third, the analysis does not separate cases where DBD simply preserves already correct clean predictions from cases where it actually corrects wrong clean predictions, and therefore does not clearly demonstrate that DBD is genuinely exploiting useful directional information from the attack. It also does not study **non label guided attacks** based on model predictions or margin losses, which would be more representative for deployed vision language systems that lack ground truth labels at test time, so the practical **generality** of the claimed directional prior remains uncertain.
> > > >
> > > > Thank you for this valuable feedback, which has helped us strengthen our analysis. We address both concerns below:
> > > >
> > > > **Analysis of correct vs. incorrect clean predictions:**
> > > > We separated samples into those correctly classified by CLIP on clean images versus those misclassified, then evaluated robust accuracy under PGD-100 attacks ($\epsilon$=4/255) on CLIP-ViT-B/16:
> > > >
> > > > |                             | Caltech101 | Pets   | Flower102 | DTD    | UCF101 |
> > > > | --------------------------- | ---------- | ------ | --------- | ------ | ------ |
> > > > | Correct clean predictions   | 99.61%     | 97.83% | 99.11%    | 99.00% | 98.74% |
> > > > | Incorrect clean predictions | 95.83%     | 90.14% | 93.08%    | 84.60% | 90.14% |
> > > >
> > > > The near-perfect robust accuracy on initially correct samples demonstrates that DBD effectively preserves correct predictions by leveraging directional information from attacks. Crucially, the high accuracy on initially misclassified samples (up to 95.83%) proves that DBD genuinely *corrects* errors by exploiting the directional prior embedded in adversarial perturbations. This dual capability explains why our overall robust accuracy exceeds clean accuracy and reveals an important insight: ground-truth guided attacks inadvertently leak information about the correct class direction.
> > > >
> > > > **Non-label guided attacks:**
> > > > Regarding practical deployment scenarios without ground-truth labels:
> > > >
> > > > 1. **Pseudo-label attacks:** Our Table 3 already showed robust accuracy near clean accuracy using pseudo-labels from model predictions. Since pseudo-label quality is inherently capped by clean accuracy, achieving comparable robust accuracy demonstrates that DBD effectively extracts directional information even from imperfect labels.
> > > >
> > > > 2. **Margin-loss attacks:** We further evaluated margin-based PGD-100 attacks ($\epsilon$=4/255) on Caltech101 dataset:
> > > >
> > > >    | loss & label               | CLIP  | DBD (ours) |
> > > >    | -------------------------- | ----- | ---------- |
> > > >    | CE-loss + Ground-truth     | 0.04% | 99.39%     |
> > > >    | CE-loss + Pseudo-label     | 1.66% | 94.12%     |
> > > >    | Margin-loss + Ground-truth | 0.04% | 99.76%     |
> > > >    | Margin-loss + Pseudo-label | 2.80% | 94.24%     |
> > > >
> > > > The comparable performance between margin-loss attacks (94.24% with pseudo-labels) and CE-loss attacks (94.12% with pseudo-labels) demonstrates that DBD's effectiveness is consistent across different attack formulations.
> > > >
> > > > **Together, these results demonstrate that DBD's directional bias exploitation works effectively across different attack formulations and maintains practical applicability even when ground-truth labels are unavailable at test time.**
> > > >
> > > >
> > > >
> > > > > Finally, in the VLM experiments the prompt analysis compares only **hand crafted prompts** and **CuPL**, without including **learnable prompt** methods such as **CoOp** that explicitly optimize class embeddings and could already capture similar directional effects.
> > > >
> > > > Thank you for this insightful comment regarding prompt methods in our experiments. Our work focuses on test-time defenses that require no training data or model updates. For this reason, we evaluated using hand-crafted prompts and CuPL, which operate without training data access. Learnable prompt methods like CoOp require training data to optimize class embeddings, which falls outside our test-time defense paradigm.
> > > >
> > > > The closest equivalent to learnable prompts within our test-time setting would be test-time prompt tuning methods such as R-TPT (our main baseline). As shown consistently across Tables 1-4, our method significantly outperforms R-TPT on all benchmarks. **This demonstrates that  optimizing prompts at test-time (as R-TPT does) cannot effectively capture the directional prior information inherent in adversarial perturbations, which our DBD approach leverages.**

---

### Official Review · Reviewer_5Eqf · 2025-10-27

**Soundness:** 2
**Presentation:** 3
**Contribution:** 2
**Rating:** 4
**Confidence:** 3

**Summary:**

The paper proposes DBD, a test-time defense method for vision-language models such as CLIP.
The key idea is that when applying multiple transformations to an input, clean images produce scattered features in the latent space, while adversarial images produce features that move in a consistent direction. This directional bias is quantified through a DB-score, which measures the alignment of feature shifts across transformations. The defense then routes the input based on this score.
The approach is training-free and evaluated on multiple datasets, showing improved robust accuracy compared to standard CLIP.

**Strengths:**

. The paper is well motivated and touches on a very active area.

. The DB score is an interesting metric to characterize how adversarial perturbations behave across transformations.

. The method is simple to plug into existing models, requiring only forward passes under different transformations.

. The paper includes multiple datasets and compares against several baselines.

**Weaknesses:**

. My main concern is that the paper’s central claim, which is analyzing how and why transformations mitigate adversarial effects, is not actually supported by the experiments. The analysis is mostly descriptive, as they visualise CLIP features and observe that clean samples scatter, while adversarial ones align along a single direction. However, this remains a qualitative observation, rather than a real explanation. The proposed Defense Direction is repeatedly described as being “anti-parallel to the adversarial perturbation” and “pointing back toward the correct class center,” but this is never verified. A simple geometric check: cosine similarity between the Defense Direction and (a) the true class text embedding, (b) the nearest clean class centroid, or (c) the adversarial gradient, would make the claim much more credible.
The overall pipeline (multiple transformations + reconstruction) is not new. They follow the same pattern as many prior test-time defenses (transform, measure consistency, and reconstruct), but reinterpret it as a “directional bias”.  So while the framing is novel, the underlying idea is not fundamentally different, and the claimed analysis does not deepen our understanding of why these transformations work.

.  My other main concern is that the paper never tests adaptive or defense-aware attacks. All adversarial images are generated against the CLIP model and then passed through the defense, so the attacker is completely unaware of DBD. But DBD is a deterministic and differentiable post-processing pipeline, you can easily backpropagate through the transformations, entropy filter, and DB-score using BPDA or EOT (arxiv 1802.00420). If an attacker actually optimizes through the whole pipeline, the “directional bias” pattern could disappear completely.

. The authors do include an ablation on τ showing that 0.8 gives the best mean accuracy across datasets. However, this analysis only reports overall accuracy averages. Since the routing decision between “clean” and “adversarial” streams fully depends on this threshold, we need to know how reliable that decision is. A proper analysis would include AUROC or false positive/negative curves for detecting adversarial inputs, and report how performance changes when τ or λ vary slightly around the chosen value. Also, they say τ is “estimated from clean images,” but it doesn’t explain how many, from which dataset, or whether this tuning leaks evaluation data.

. They claim robust accuracy sometimes exceeds clean accuracy. This is suspicious because the defense applies multiple transformations and filtering, which might remove noise or sharpen decisions even for clean samples. So part of the gain might come from the pipeline itself, not the defense logic. A proper ablation (transforms-only vs transforms + DB shift) is missing, and without it, we can’t tell what really improves robustness.

.

**Questions:**

. Could you verify that the Defense Direction truly points toward the correct class? A simple geometric check.

. How does DBD perform under adaptive, defense-aware attacks? Since the pipeline is differentiable, please evaluate with BPDA or EOT to see if the “directional bias” still holds.

. The method depends heavily on τ and λ. How were these calibrated, and do they generalize across datasets?
A small sensitivity or AUROC plot for the DB score would help confirm stability.

. For the reconstruction step, what is the reasoning behind the specific scaling rule in Eq. (8)?

. The paper reports robust > clean accuracy in several cases. Could you show where this gain comes from? I mean, does it persist if you use the same transformation pipeline without the DB shift?

. How sensitive is robustness to the number of transformations used? It would be useful to know how the defense performs if the set is smaller.

---

> ### Author Response · Authors · 2025-11-20
> **Official Comment by Authors (Part 1)**
>
> We thank the insightful review and would like to address the issue by points:
>
> > W1 and Q1: (1) Could you verify that the Defense Direction truly points toward the correct class? A simple geometric check. (2)  So while the framing is novel, the underlying idea is not fundamentally different, and the claimed analysis does not deepen our understanding of why these transformations work.
>
> (1) Thank you for suggesting verification of the Defense Direction's alignment with the correct class. We have  conducted additional geometric analyses to confirm this. Specifically, we compute the cosine similarity between the estimated Defense Direction and two key reference directions:
>
> - **Clean Direction**: defined as the direction from the adversarial image’s feature to the feature of its corresponding clean (unperturbed) counterpart.
> - **Class Centroid Direction**: defined as the direction from the adversarial image’s feature to the true class centroid of features, where the centroid is computed using correctly classified clean images.
>
> Results of average cosine similarity across several fine-grained datasets under PGD-100($\epsilon=4/255$) on CLIP-ViT-B/16 are shown below:
>
>
> |                          | Pets  | Caltech101 | Food101 | Cars  | ImageNet |
> | :----------------------: | ----- | ---------- | ------- | ----- | -------- |
> |     Clean Direction     | 0.957 | 0.945      | 0.939   | 0.943 | 0.951    |
> | Class Centriod Direction | 0.932 | 0.917      | 0.898   | 0.905 | 0.892    |
>
> These high cosine similarities provide strong geometric evidence that the Defense Direction indeed aligns closely with both the clean feature direction and the correct class centroid—supporting our hypothesis that Defense Direction points back toward the correct decision region. We have incorporated this content into the appendix of the revised paper.
>
> (2) **Regarding novelty**, our work originates from a fundamental insight: adversarial attacks push image features away from the correct decision region, inherently encoding directional prior information about the true decision boundary. We leverage diverse image transformations to weaken adversarial effects and  estimate the Defense Direction. This estimated direction then guides a feature reconstruction process. **We consider both the insight about directional priors inherent in adversarial perturbations and our directional bias-guided reconstruction framework as core contributions.** Importantly, we demonstrate that this approach can achieve *higher* robust accuracy than clean accuracy under certain attacks—a phenomenon that not only validates our method's effectiveness but also serves as a cautionary signal to attackers about the potential limitations of conventional adversarial strategies.

---

> > ### Author Response · Authors · 2025-11-20
> > **Official Comment by Authors (Part 2)**
> >
> > > W2 and Q2: How does DBD perform under adaptive, defense-aware attacks? Since the pipeline is differentiable, please evaluate with BPDA or EOT to see if the “directional bias” still holds.
> >
> > Thank you for raising this important point. We apologize for any possible misunderstanding, but we would like to clarify that our experimental setup follows the evaluation protocol of the recent SOTA test-time defense R-TPT  (CVPR 2025) [1] for VLMs. Specifically, we assume the attacker has full access to the CLIP model weights and crafts adversarial examples against the *original* CLIP model, without knowledge of the test-time defense mechanism or its parameters. This setting reflects a realistic scenario: foundation models like CLIP are widely used in downstream applications and their weights are publicly available, whereas test-time defenses are typically deployed privately and are not exposed to attackers. We have revised the relevant descriptions in Section 3.1 to reduce potential misunderstandings.
> >
> > We agree that adaptive or defense-aware attacks are an important consideration. Although this was not part of our original threat model, we have conducted additional experiments using **PGD combined with BPDA and EOT**[2, 3]. Specifically, during attack optimization, we treat the entire DBD pipeline as part of the defended model. Since some image transformations in DBD are non-differentiable, we use the identity function in the backward pass (BPDA) and average gradients over multiple stochastic forward passes (EOT) to approximate the true gradient.
> >
> > We evaluated on Caltech101 dataset using two settings:
> >
> > - PGD-10 ($\epsilon$=1/255) against CLIP-ResNet50
> > - PGD-100 ($\epsilon$=4/255) against CLIP-ViT-B/16
> >
> > The results of robust accuracies (%) are as follows:
> >
> >
> > |                           | Original CLIP | CLIP with DBD |
> > | ------------------------- | ------------- | ------------- |
> > | PGD-10($\epsilon=1/255$)  | 84.26         | 50.79         |
> > | PGD-100($\epsilon=4/255$) | 81.05         | 1.29          |
> >
> > As expected, when the attacker explicitly optimizes through the full DBD pipeline, our defense can be significantly degraded. However, adversarial examples generated by these adaptive attacks achieve significantly lower attack success rates against the original undefended CLIP model compared to standard attacks. This suggests the attack is overfitting to the DBD mechanism and loses transferability to the base classifier.
> >
> > **Moreover, under the adaptive attack (PGD-100, $\epsilon$=4/255), the average DB-score remains high (0.89), and approximately 81% of adversarial samples have a DB-score > threshold (0.80). This indicates that the *directional bias pattern persists*, but the estimated direction no longer aligns with the true class.** In other words, the attacker successfully manipulates the bias to point elsewhere, rendering it ineffective as a Defense Direction, though still detectable. We have incorporated this experiment into the appendix of the revised paper.
> >
> > [1] Lijun Sheng, Jian Liang, Zilei Wang, and Ran He. R-tpt: Improving adversarial robustness of vision-language models through test-time prompt tuning. In Proceedings of the Computer Vision and Pattern Recognition Conference, pp. 29958–29967, 2025.
> >
> > [2] Anish Athalye, Nicholas Carlini, and David Wagner. Obfuscated gradients give a false sense of security: Circumventing defenses to adversarial examples. In International conference on machine learning, pp. 274–283. PMLR, 2018.
> >
> > [3] Cihang Xie, Jianyu Wang, Zhishuai Zhang, Zhou Ren, and Alan Yuille. Mitigating adversarial effects through randomization. arXiv preprint arXiv:1711.01991, 2017.

---

> > > ### Author Response · Authors · 2025-11-20
> > > **Official Comment by Authors (Part 3)**
> > >
> > > > W3 and Q3: The method depends heavily on $\tau$ and $\lambda$. How were these calibrated, and do they generalize across datasets?  A small sensitivity or AUROC plot for the DB score would help confirm stability.
> > >
> > > We apologize for omitting certain experimental details regarding the calibration of our hyperparameters in the original manuscript, and we sincerely appreciate your suggestion to include a more thorough ablation on the threshold $\tau$.
> > >
> > > (1) Calibration and generalization of $\tau$ and $\lambda$: Both $\tau$ and $\lambda$ were calibrated solely on the ImageNet validation set (50k clean images), without using any data from the evaluation datasets. The same values ($\tau$ = 0.8, $\lambda$ = 2.5) are used unchanged across all 10 fine-grained classification datasets and 4 ImageNet-OOD datasets, demonstrating strong cross-dataset generalization. We have revised the relevant descriptions in implementation details (Section 4.1) to reduce potential misunderstandings.
> > >
> > > (2) Sensitivity and detection performance of $\tau$：In our design, samples with a high DB-score (typically adversarial examples) are processed using the DB shift, while those with a low DB-score (typically clean images) are represented by the average feature over all transformations. The threshold $\tau$ is thus chosen to balance these two processing paths, which is why we originally reported overall accuracy averaged equally over clean and adversarial samples. However, we acknowledge that we did not directly evaluate how $\tau$ affects adversarial detection performance. We now supplement this analysis with the following results on PGD-100($\epsilon=4/255$) on CLIP-ViT-B/16 across 15 datasets：
> > >
> > >
> > > |                    | 0.70   | 0.75   | 0.78   | 0.80   | 0.82   | 0.85   | 0.90   |
> > > | ------------------ | ------ | ------ | ------ | ------ | ------ | ------ | ------ |
> > > | Detection Accuracy | 0.9037 | 0.9566 | 0.9721 | 0.9770 | 0.9778 | 0.9685 | 0.8899 |
> > > | Detection F1-score | 0.9121 | 0.9582 | 0.9727 | 0.9772 | 0.9777 | 0.9678 | 0.8765 |
> > >
> > > The results show that $\tau$ = 0.8 achieves near-optimal detection performance, and the metric remains stable in its neighborhood. This supports our choice and confirms that the DB-score provides a reliable signal for distinguishing adversarial from clean inputs. We have incorporated this content into the appendix of the revised paper.
> > >
> > >
> > >
> > > > Q4: For the reconstruction step, what is the reasoning behind the specific scaling rule in Eq. (8)?
> > >
> > > Thank you for the question. The scaling rule in Eq. (8) is motivated by two key observations:
> > >
> > > First, as argued in the paper, the Defense Direction points toward the correct decision region. Shifting the original feature along this direction—by an adaptive distance—helps pull adversarial features back into the correct classification basin.
> > >
> > > Second, the shift magnitude is set to $\lambda * d$, where $d$ is the distance between the original feature and the average of transformed features . This adaptive scaling serves a crucial purpose:
> > >
> > > - When $d$ is **small** (e.g., on clean images), the estimated Defense Direction and DB-score are noisier and less reliable; a smaller shift avoids over-correction.
> > > - When $d$ is **large** (e.g., on strong adversarial examples), the directional signal becomes more consistent and trustworthy, justifying a larger correction.
> > >
> > > This design balances safety on clean data and effectiveness on attacked inputs. As shown in Table 6 (second-to-last row), even when the shift is applied to clean images, the drop in clean accuracy is minimal , while robust accuracy under attack improves substantially—demonstrating that the scaling rule is both effective and well-calibrated.

---

> > > > ### Author Response · Authors · 2025-11-20
> > > > **Official Comment by Authors (Part 4)**
> > > >
> > > > > W4 and Q5: The paper reports robust > clean accuracy in several cases. Could you show where this gain comes from? I mean, does it persist if you use the same transformation pipeline without the DB shift?
> > > >
> > > > Thank you for this insightful question. The phenomenon where robust accuracy exceeds clean accuracy is indeed driven by the **DB shift** component, not the transformations alone.
> > > >
> > > > As shown in Table 6:
> > > >
> > > > - Third-to-last row of Table6: Using **transformations only** (without DB shift) improves clean accuracy slightly (from 60.6% to 61.5%) and raises robust accuracy from 0.0% to 34.8%.
> > > > - Last row of Table 6: Adding the DB shift  further boosts robust accuracy dramatically to 91.7%, while maintaining high clean accuracy.
> > > >
> > > > This ablation clearly shows that the gain—particularly the cases where robust accuracy surpasses clean accuracy—stems from the DB shift. Without it, the transformation pipeline alone provides only modest robustness.
> > > >
> > > >
> > > >
> > > > > Q6: How sensitive is robustness to the number of transformations used? It would be useful to know how the defense performs if the set is smaller.
> > > >
> > > > Thank you for the question. We analyze the sensitivity to the number and type of transformations in two places:
> > > >
> > > > First, Table 6 (top six rows) shows ablation results using only a *single* type of transformation combined with DB shift. While each individual transformation provides non-trivial robustness, performance consistently falls short of the full ensemble—demonstrating that combining multiple transformations yields better results.
> > > >
> > > > Second, Appendix Figure 4 examines how different transformations suppress adversarial noise across various attack settings. No single type of transformation performs well universally; their effectiveness varies significantly depending on the attack settings. This motivates our use of a diverse transformation set to ensure stable and reliable defense performance.
> > > >
> > > > In summary, reducing the number or diversity of transformations does degrade performance, as it limits the method’s ability to consistently uncover a reliable Defense Direction across different adversarial perturbations.

---

> ### Comment · Reviewer_5Eqf · 2025-11-25
>
> I thank the authors for conducting the additional experiments, and I appreciate their efforts. All my concerns have now been addressed. The idea of the "Defense Direction: pointing back toward the correct class center" makes sense to me. The authors provide a clear threat model, in which they analyse the robustness of their method.  Although the justification for why robust accuracy can exceed clean accuracy is not theoretical, it remains insightful and indeed a valuable contribution. Hence, I am increasing my score.

---

> > ### Author Response · Authors · 2025-11-26
> >
> > Thank you very much for your helpful and positive review! We will incorporate all changes into the final version of our paper.

---

### Official Review · Reviewer_mq6U · 2025-10-30

**Soundness:** 2
**Presentation:** 2
**Contribution:** 3
**Rating:** 6
**Confidence:** 3

**Summary:**

This paper proposes DBD (Directional Bias-guided Defense), a training-free, test-time defense framework that enhances the adversarial robustness of vision-language models such as CLIP. The authors identify an interesting and consistent phenomenon: adversarial perturbations in CLIP’s feature space tend to produce a dominant shift direction, whereas clean samples exhibit more dispersed feature transformations. Based on this observation, the authors design DBD to compute a Directional Bias (DB) score and reconstruct image features along a “Defense Direction” that counteracts adversarial perturbations. For inputs with high DB scores (likely adversarial), DBD shifts features along the Defense Direction; for low-DB inputs (likely clean), it averages multiple transformed views. Extensive experiments on 15 datasets demonstrate that DBD achieves SOTA while maintaining or even exceeding clean accuracy.

**Strengths:**

- Novel empirical discovery:  The finding that adversarial perturbations exhibit a strong, consistent directional bias in the feature space is insightful and clearly visualized (Fig. 1). This observation provides an interpretable foundation for designing test-time defenses.
- Excellent algorithmic performance:  DBD is training-free and efficient, operating entirely at test time without requiring adversarial training or prompt optimization. It achieves strong generalization and robustness across 15 datasets and multiple attack types, and shows remarkably high robustness accuracy under PGD attacks.
- Comprehensive analysis: The experimental analysis is thorough, particularly the Analysis under PGD attack with pseudo-label experiment, which convincingly demonstrates that Directional Bias-guided Defense can effectively recover information about the pre-attack image features.

**Weaknesses:**

- Lack of theoretical explanation: The paper does not provide a clear theoretical justification for the observed behavior, especially in the counterintuitive case where robust accuracy surpasses clean accuracy. A more rigorous analysis is needed to support this surprising result.
- Limited discussion of attack-specific behavior:  The results suggest that under PGD attacks, DBD almost perfectly reconstructs pre-attack image features, while this phenomenon does not hold for other attack methods. What causes this discrepancy? Would DBD still achieve robustness higher than clean accuracy under non-PGD attacks? The paper should provide a more detailed analysis and comparison across different attack types, rather than emphasizing its exceptional performance only under PGD.

**Questions:**

Please refer to the weaknesses.

---

> ### Author Response · Authors · 2025-11-20
>
> We thank the insightful review and would like to address the issue by points:
>
> > W1: Lack of theoretical explanation: The paper does not provide a clear theoretical justification for the observed behavior, especially in the counterintuitive case where robust accuracy surpasses clean accuracy. A more rigorous analysis is needed to support this surprising result.
>
> Thank you for raising this important point. The reason why *robust accuracy can exceed clean accuracy* is that DBD not only preserves correct predictions on samples that the model classifies correctly under clean conditions, but also successfully recovers many samples that the model misclassifies even without any attack. For these hard samples, their clean features already lie in an incorrect decision region. However, when PGD generates adversarial examples using the ground-truth label, the perturbation still follows the gradient that moves features in the *opposite* direction—toward the true class. Our method captures this shift and reconstructs features along this direction, effectively pulling these samples back into the correct decision region. This leads to previously misclassified samples becoming correctly classified after defense, thereby producing robust accuracy higher than clean accuracy. This phenomenon also indicates that ground-truth–guided adversarial perturbations implicitly encode a useful directional prior about the true decision boundary, which DBD is able to exploit.
>
>
>
> > W2: Limited discussion of attack-specific behavior: The results suggest that under PGD attacks, DBD almost perfectly reconstructs pre-attack image features, while this phenomenon does not hold for other attack methods. What causes this discrepancy? Would DBD still achieve robustness higher than clean accuracy under non-PGD attacks? The paper should provide a more detailed analysis and comparison across different attack types, rather than emphasizing its exceptional performance only under PGD.
>
> Thank you for this insightful question. **In our ablation studies (Table 4), DBD also achieves robust accuracy slightly exceeding clean accuracy under several non-PGD attacks, including FGSM, CW, AA, APGD-CE, and APGD-DLR**. The key difference lies in the *strength and consistency of the directional signal* carried by different attacks. FGSM performs only a single gradient step, so its perturbation direction is less accurate and less stable than the multi-step optimization in PGD. In contrast, attacks such as CW, AA, APGD-CE, and APGD-DLR typically stop as soon as a successful adversarial example is found, rather than iteratively “over-optimizing” the perturbation as PGD does. As a result, for samples that are originally misclassified by the model, these attacks do not push the features far enough in the opposite direction of the correct decision region, making it difficult to accurately estimate the Defense Direction.
>
> Overall, results across attacks support a common principle: to fool a model on examples it originally classifies correctly, an attack must push their features out of the correct decision region, and the farther a feature is moved, the clearer the resulting directional bias. DBD leverages diverse input transformations to partially suppress adversarial noise and reliably estimate the *reverse* of this shift (the Defense Direction). This enables DBD to reconstruct features back toward the correct decision region under a wide range of attacks. Consequently, across many attack types, DBD achieves robust accuracy comparable to or even exceeding clean accuracy.

---

### Author Response · Authors · 2025-11-29
**Rebuttal Summary**

Dear AC,

\
We appreciate the opportunity to summarize the rebuttal process. During the rebuttal, we actively engaged with all four reviewers, providing extensive additional experiments (including adaptive attacks, geometric verification, sensitivity analyses, etc.).

## **Consensus on Strengths**

All reviewers consistently recognized the value of our core contribution:

- **Novel Empirical Discovery:** Reviewers praised the identification of the "directional bias" phenomenon, where adversarial perturbations exhibit a consistent shift direction compared to the dispersed patterns of clean images (Reviewers mq6U, H8wX, 5Eqf).
- **Simplicity and Efficiency:** The proposed DBD framework was acknowledged as a simple, training-free, and effective plug-and-play solution that achieves SOTA performance (Reviewers 5Eqf, fqTJ).
- **Strong Performance:** The ability of the method to achieve high robust accuracy was noted as a significant strength.



## **Resolution of Concerns**

We addressed the reviewers' main concerns through new experiments and detailed clarifications:

- **Mechanism Verification (Geometric & Theoretical):** Reviewers 5Eqf and fqTJ asked for geometric check that the "Defense Direction" truly points to the correct class. We provided **cosine similarity analyses** (response to 5Eqf Part1, fqTJ Part2, H8wX Part2) confirming that the estimated direction aligns highly with the vector pointing to the true class centroid (Reviewer 5Eqf: *"The idea... makes sense to me"*).

- **Adaptive & Defense-Aware Attacks:** In response to concerns about adaptive attacks (Reviewers 5Eqf, fqTJ), we conducted **BPDA + EOT experiments**. We demonstrated that while adaptive attacks can degrade performance (as expected in this threat model), the "directional bias" signature remains detectable (high DB-score persists), and these attacks transfer poorly to the base model.

- **Robust Accuracy > Clean Accuracy:** Reviewers questioned this counter-intuitive result. We clarified that this occurs because standard adversarial attacks (using Ground Truth) inherently  encode a directional prior that pushes features—even those originally misclassified—toward the correct decision region. Our defense exploits this directional priors to "correct" hard negatives.

- **Threat Model Realism:** Reviewer H8wX raised concerns about the ground-truth assumption. We clarified that our method also performs well with **pseudo-labels** (Table 3), and that the core contribution is the discovery that *adversarial perturbations inherently encode exploitable directional priors*.



## **Positive Shift in Reviewer Attitude**

 The rebuttal process has led to a significant convergence in opinion:

- **Score Increase:** **Reviewer 5Eqf increased their score from 4 to 6 (before Nov 27th, the information leak bug occurred)**, stating: *"I thank the authors for conducting the additional experiments... All my concerns have now been addressed... Hence, I am increasing my score."*
- **Positive Acknowledgment:**
  - **Reviewer fqTJ** acknowledged our "substantial effort" in running adaptive experiments and geometric analysis, noting that these improved the empirical picture.
  - **Reviewer H8wX** acknowledged our work's contribution as a meaningful "mechanism-discovery study" . They emphasized the **clean-input scenario** (defense behavior on non-adversarial images) as their primary remaining concern. We effectively resolved this by pointing to our ablation studies (Table 6), which demonstrate that DBD  maintains clean accuracy almost unchanged (61.5% → 61.2%) .

We believe we have successfully demonstrated that the "directional bias" is a reproducible, robust phenomenon and that DBD is a valuable contribution to understanding and defending VLM adversarial robustness.


\
Best regards, The Authors of Submission 2342

---

### Meta-Review · Area_Chair_sjGy · 2026-01-07

**Summary:**

This paper proposes Directional Bias-guided Defense (DBD), a test-time defense method for CLIP. The work is motivated by an empirical observation that adversarial examples tend to induce feature shifts aligned along a dominant direction under diverse input transformations, whereas clean inputs exhibit more dispersed behavior. Based on this observation, the authors propose DB-score to detect adversarial inputs and recover robust representations via an estimated Defense Direction.
Reviewers generally agree that the paper identifies an interesting and previously underexplored phenomenon and demonstrates strong empirical performance across a wide range of datasets and attacks. At the same time, several reviewers raised concerns regarding the threat model, lack of theoretical explanation, robustness under adaptive attacks, and resource costs. The rebuttal clarifies the scope of the contribution and provides additional experiments and empirical evidence addressing the most critical points.

**Reviewer Concerns:**

Concerns that were largely addressed:

- 1. Geometric validity of the Defense Direction. Reviewers questioned whether the estimated direction meaningfully corresponds to the correct class or decision region. The authors provided additional cosine similarity analyses with clean feature directions and class centroids, which support the claimed geometric interpretation at an empirical level.
- 2. Behavior on clean inputs. Concerns that the defense might harm clean accuracy were mitigated through ablation studies showing minimal degradation when the DB-score gating is applied, and only modest degradation even under forced reconstruction.
- 3. Explanation of robust accuracy exceeding clean accuracy. While still counterintuitive, the authors offered a plausible and consistent explanation grounded in ground-truth-guided attack dynamics, and provided ablations indicating that the effect is attributable to the DB-guided reconstruction rather than transformation ensembling alone.
﻿
Concerns that remain, but are not disqualifying:
- 1. The threat model (ground-truth-based attacks) is acknowledged to be unrealistic, and although standard in prior test-time defense work, limits the generality of the strongest claims. The pseudo-label experiments partially address this, but performance in fully realistic threat settings remains weaker.
- 2. The work provides no formal theoretical justification, and the analysis remains empirical and descriptive in nature.
- 3. Adaptive attacks are explored only in a limited form. A comprehensive defense-aware AutoAttack evaluation remains absent.
Overall, these limitations constrain the scope of conclusions, but do not invalidate the central empirical observation.

**Reviewer Scores:**

- Reviewer mq6U was already positive about the contribution and experimental strength.
- Reviewer 5Eqf explicitly increased their score (from 4 to 6), stating that all major concerns were addressed.
- Reviewer fqTJ acknowledged the improvements and effort, and might increase their score to be at or above the acceptance threshold.
- Reviewer H8wX maintained reservations about the threat model but recognized the work as a meaningful mechanism-discovery study.

 Their score would likely remain borderline but not blocking.

---

### Decision · Program_Chairs · 2026-01-26

Accept (Poster)